# Decision uncertainty as a context for motor memory

**Kisho Ogasa** [1], **Atsushi Yokoi** [1,2], **Gouki Okazawa** [3], **Morimichi Nishigaki** [4], **Masaya Hirashima** [1,2] & **Nobuhiro Hagura** [1,2]✉

The current view of perceptual decision-making suggests that once a decision is made, only a single motor programme associated with the decision is carried out, irrespective of the uncertainty involved in decision making. In contrast, we show that multiple motor programmes can be acquired on the basis of the preceding uncertainty of the decision, indicating that decision uncertainty functions as a contextual cue for motor memory. The actions learned after making certain (uncertain) decisions are only partially transferred to uncertain (certain) decisions. Participants were able to form distinct motor memories for the same movement on the basis of the preceding decision uncertainty. Crucially, this contextual effect generalizes to novel stimuli with matched uncertainty levels, demonstrating that decision uncertainty is itself a contextual cue. These findings broaden the understanding of contextual inference in motor memory, emphasizing that it extends beyond direct motor control cues to encompass the decision-making process.

In the penalty shootout of a football (soccer) game, a player may decide to confidently kick the ball to the right corner, seeing that the goal-keeper is moving to the other side, or he may alternatively decide to make the same kick while being unsure about the goalkeeper's movement. Because both decisions lead to 'apparently' identical actions, we believe that the same motor memory (a motor programme for kicking the ball to the right) is retrieved and executed for both cases, regardless of the quality of the preceding decision. But is this true?

In perceptual decision-making studies, uncertainty is treated as a factor that modulates the decision-making process. Perceptual decision-making is studied as an evidence-accumulation process in which the decision is terminated when the accumulated evidence in favour of one of the options reaches a predetermined threshold (decision-bound)[1–3]. Here, uncertainty determines how quickly evidence accumulates; however, once a decision has been made, it is implicitly assumed that the subsequent action does not consider the preceding uncertainty.

Dominant theories of motor learning posit that the brain flexibly forms and switches between multiple motor memories through contextual inference processes, relying on external sensory cues that directly specify an action (for example, in ref. 4). Such cues include the visual appearance of an object (tools), the type and/or location of the (future) reached targets[5–7] and the posture/state of other body parts during an action[8–10]. Therefore, contextual cues for motor memories are assumed to be limited to factors directly relevant to motor execution and do not include the state of the decision preceding the action.

However, learning to perform an action differently on the basis of the uncertainty of a decision seems sensible, as subjective uncertainty can be correlated with important behavioural factors, including the expected outcome of an action or the possibility of revising a motor plan[11,12]. An action following certain decisions may allow people to plan the next move ahead, given the high probability of making a correct decision. Action during uncertain decisions may restrict such sequential planning, given the possibility of decision failure.

Contrary to the dominant views in both decision-making and motor-learning literature, we show here that decision uncertainty can work as a contextual cue for motor memory. In other words, we

¹Center for Information and Neural Networks (CiNet), National Institute of Information and Communications Technology (NICT), Osaka, Japan. ²Graduate School of Frontier Biosciences, Osaka University, Osaka, Japan. ³Institute of Neuroscience, Key Laboratory of Primate Neurobiology, Center for Excellence in Brain Science and Intelligence Technology, Chinese Academy of Sciences, Shanghai, China. ⁴Innovative Research Excellence, Honda R&D Co. Ltd, Utsunomiya, Japan. ✉e-mail: n.hagura@nict.go.jp

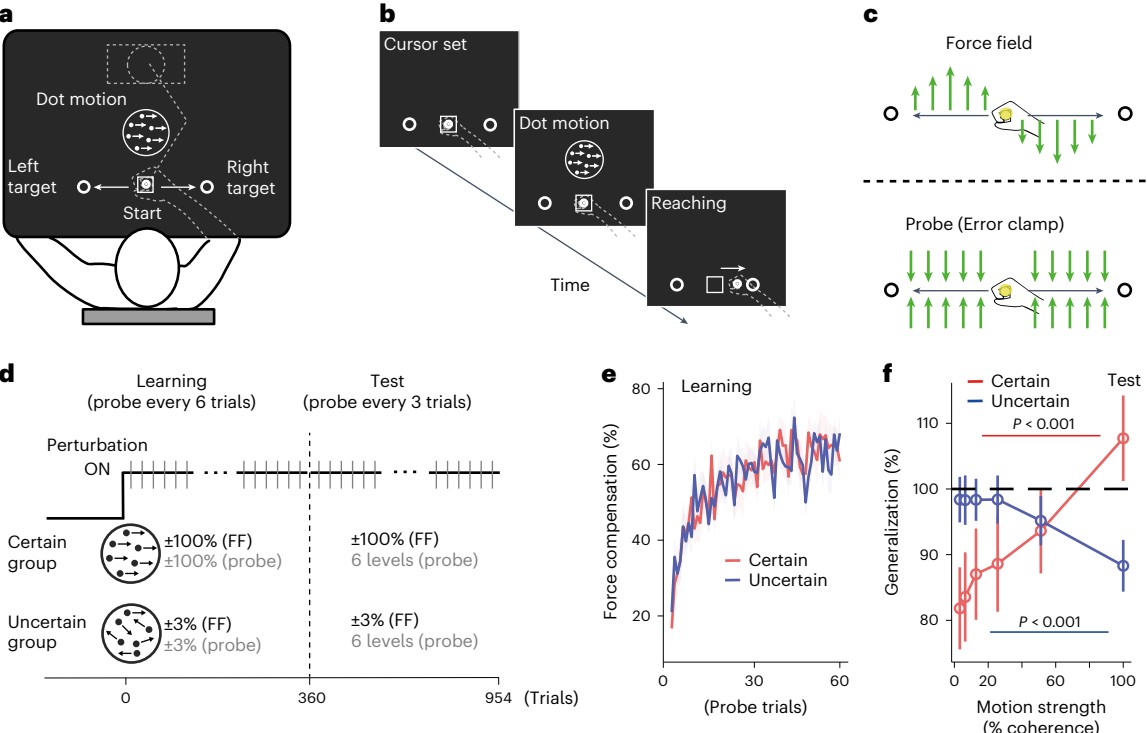

**Fig. 1 | Decision uncertainty-level-dependent tuning of motor memory (Exp. 1). a,b,** Target-reaching task preceded by motion discrimination decision. The participants held a handle of a manipulandum and judged the direction of the random-dot motion. The decision was made by moving a handle towards the target in the direction of the perceived motion. **c,d,** Force-field learning under different uncertainties. In the learning phase (**d** left), participants learned to make a straight reaching movement towards the target by resisting a velocity-dependent force field (green arrows, **c** top). One group learned the force following the decision for a certain stimulus (certain-decision group (*n* = 19): 100% coherent motion), whereas the other group learned the force following an uncertain stimulus (uncertain-decision group (*n* = 19): 3.2% coherent motion). Occasionally, probe trials were presented (green arrows in **c** bottom, grey lines in **d**), in which the movement trajectory of the hand was constrained to a straight path (error clamp). The level of motor memory retrieval was measured as the

force produced against this virtual wall. In the test phase (**d** right), participants performed the same force-field task as in the learning phase, but the probe trials included motion with six different uncertainty levels for the left and right sides (±3.2%, 6.4%, 12.8%, 25.6%, 51.2% and 100%; positive values indicate rightward motion). **e,** Progression of force-field learning (probe error-clamp trials) during the learning phase. The vertical axis indicates the force-field compensation level and a value of 100 indicates full compensation of the perturbation. **f,** Generalization of motor memory across different uncertainty levels during the test phase. The vertical axis indicates the generalization of learned memory across different uncertainty levels, calculated as the ratio of the force-field compensation level during the test phase to the final block of the learning phase (two-sided paired *t*-test). Error bars and shading indicate s.e.m. across participants.

demonstrate that specific motor memories can be formed on the basis of the level of decision uncertainty preceding an action. The findings of this study broaden our understanding of the contextual inference for motor memory by extending the scope of contextual cues from direct sensory input associated with motor control to the state of the decision-making process.

## Results

### Motor memory tuned to the level of decision uncertainty

Previous studies on episodic memory have established that the shared context between learning and retrieval facilitates successful memory recall[13]. Therefore, to examine the dependency of motor memory on the decision uncertainty context, we first tested whether the action learned following a particular decision uncertainty could be retrieved better when a decision with the same uncertainty level preceded it.

Participants (*N* = 38) judged the direction (left or right) of a visual random-dot motion stimulus presented on a screen (Fig. 1a,b). Participants were assigned to one of two groups: the certain-decision group (*n* = 19) judged the direction of 100% coherent random-dot motion, while the uncertain-decision group (*n* = 19) judged the direction of 3.2% coherent motion. In the certain-decision group, because the direction of motion was easy to decide, the participants were able to decide the direction quickly and with high confidence. In contrast, the

confidence level was lower in the uncertain-decision group, and the decision required a longer time to initiate (Supplementary Fig. S1a,b and Fig. 4c). Following this decision, participants made a straight centre-out reaching movement towards the target in the direction of the perceived motion (Fig. 1a,b).

The experiment comprised two phases: learning and test (Fig. 1d). After familiarization with the random-dot motion decision task, each group of participants entered the learning phase, where they performed the decision-making task with a velocity-dependent curl force field (force-field trials)[14,15] applied to the reaching movement (Fig. 1c top). To account for this force-field perturbation and to make a straight reaching movement, the participants had to learn to generate an appropriate amount of perpendicular force to counteract the perturbating force. To measure this counteracting force (force compensation level), the probe trials were occasionally interleaved with the force-field trials (Fig. 1c bottom and grey lines in Fig. 1d). The trajectory of reaching during the probe trials was constrained to a straight path between the starting position and the target (channel), and the force the participants applied to the wall of the channel was measured (error-clamp trials) (Fig. 1c bottom). This allowed us to measure the amount of force (or motor memory) retrieved in that trial while avoiding the occurrence of any kinematic errors that could affect learning[16] (see Methods). During the learning phase (Fig. 1e), the force compensation

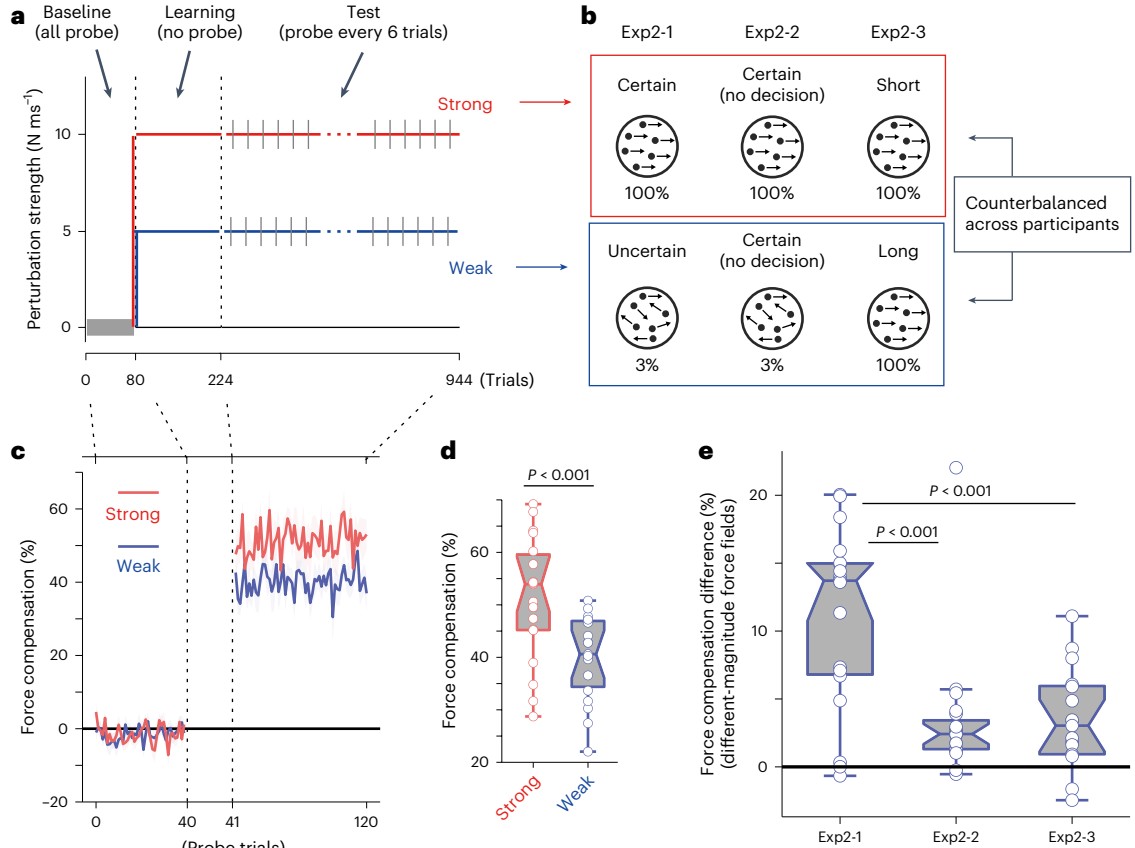

**Fig. 2 | Simultaneous learning of different-magnitude force fields based on the decision uncertainty context (Exp.2).** Results of force-field learning of two different magnitudes. **a**, Experimental trial structure. The participants learned two force-field magnitudes, each associated with a different decision uncertainty level. The grey line indicates probe error-clamp trials. **b**, Conditions of the main (Exp. 2-1) and two control (Exp. 2-2, 2-3) experiments. In Exp. 2-1 (*n* = 19), certain (100% coherent motion) and uncertain (3.2% coherent motion) decisions are associated with either strong (red) or weak (blue) force fields. In Exp. 2-2 (*n* = 20), 100% and 3.2% motion stimuli were each associated with different (strong or weak) force fields; however, the participants did not make any motion direction decisions and only reached a specified target after the stimulus disappeared. In Exp. 2-3 (*n* = 17), two different durations of motion stimuli, both with 100% motion coherence, were each associated with different force fields, and the participants reached the target of their motion direction

decision. The association between uncertainty level and force-field strength was counterbalanced across participants. **c,d**, Force compensation measured in probe error-clamp trials during Experiment 2-1 (two-sided paired *t*-test). Data were aligned according to the strength of the force field (strong or weak). The vertical axis shows force compensation with respect to a strong force. Thus, full compensation for the weak force was 50%. **e**, Comparison of the effect with control experiments, expressed as the difference in the force compensation level between the two force fields (two-sided two-sample *t*-test, corrected). Error bars and shading indicate the s.e.m. across participants. In the boxplots, each dot represents a participant, the midline of the box represents the median, box limits indicate the 25th to the 75th percentile and whiskers show the range (minimum to maximum) of the data. Outliers are determined by data points that are greater than 1.5× size from the box.

level gradually increased to 64.8 ± 18.4% (s.d.) for the certain-decision group and 63.1 ± 10.9% for the uncertain-decision group. Importantly, there was no significant difference in the force compensation between the two groups in the final learning block (12 trials) (two-sample *t*-test, $t_{(18)} = 0.35$, $P = 0.727$, effect size (dz) = 0.12). This suggests that participants in both groups successfully learned to compensate for the force field, and that learning of the force field itself was not affected by the level of preceding decision uncertainty.

In the test phase, we tested how motor memory formed following certain (100% motion coherence level; certain-decision group) and uncertain (3.2% motion coherence level; uncertain-decision group) decisions generalizes to actions following different preceding decision uncertainty levels. Decision stimuli in the force-field trials were the same as in the learning phase (100% coherent motion for the certain-decision group and 3.2% coherent motion for the uncertain-decision group), but the probe trials included six different uncertainty levels (±3.2%, 6.4%, 12.8%, 25.6%, 51.2% and 100%; a positive value indicates rightward motion). If the same motor programme is learned/executed for the same left–right decision, regardless of the

level of uncertainty, participants should exert the same counteracting force across different levels of decision uncertainty. However, this was not the case.

Participants in the certain-decision group exhibited force at the learned level (107.7% of the learning phase, mean force compensation: 65.9 ± 12.9% (s.d.)) when the decision was certain (100% coherent motion), but the force dropped to around 80% (81.9%) of the learned level (mean force compensation: 49.2 ± 12.9%) when the decision was uncertain (3.2% coherent motion, which is untrained) (paired *t*-test, $t_{(18)} = 8.68$, $P < 0.001$, dz = 1.99) (Fig. 1f, red line). Similarly, participants in the uncertain-decision group exhibited the force at the learned level (98.4% of the learning phase) (mean: 62.0 ± 12.3%) when the decision was uncertain (3.2% coherent motion), but the force again significantly dropped (88.3%) for certain decisions (100% coherent motion) (mean: 55.4 ± 12.0%) (paired *t*-test, $t_{(18)} = 6.12$, $P < 0.001$, dz = 1.40) (Fig. 1f, blue line). Thus, the manner in which the force was retrieved depended on the decision uncertainty level at which participants learned the force-field (Fig. 1f) (2-way analysis of variance (ANOVA), motion coherence (6) × group (2) interaction effect, $F_{(5,180)} = 55.736$, $P < 0.001$,

$\eta^2 = 0.61$). Such reversed retrieval patterns of force between the two groups cannot be explained by the generally deteriorated motor output following uncertain decisions[17] as this would predict that the force would drop towards higher-uncertainty decisions regardless of the different learning experiences.

The difference in the motor learning quality between the groups could not explain this result, as the rate and magnitude of motor learning were comparable between the two groups (Fig. 1e). It is also unlikely that the features of the visual stimuli (100% and 3.2% coherent motion) are the primary determinants of this effect. Previous study showed that the background colour of a workspace cannot be a contextual cue for motor memory[18]. Although the peripheral target rotation 'direction' stimulus has been shown to be a weak contextual cue for motor memory[18,19], the stimulus used in the present study had a motion coherence level independent of 'direction'. In the following experiments, we further showed that the motion coherence level itself (not decision uncertainty) cannot be a contextual cue for motor memory (see below for the results of Experiments 2-2 and 3-3).

We also fitted a drift-diffusion model (DDM) to the choice and reaction time data of the test phase to quantify the differences in the decision process between the certain and uncertain-decision groups (Methods). We found that the uncertain-decision group had slightly higher visual motion sensitivity, and their judgements were more conservative than those of the certain-decision group (Supplementary Fig. S1c,d). This was probably due to the perceptual learning induced by repeated exposure to the weak motion signal in the uncertain group, and the repetition of the difficult decision made the participants' decision more conservative[20,21]. This tendency was also observed in the pattern of the peak velocity of movement across different motion coherence levels (Supplementary Fig. S1f). However, these results still cannot directly explain the reversed force production patterns across different decision uncertainty levels between the two groups.

Taken together, the incomplete transfer of motor memory across different decision uncertainties implies that part of the motor memory is formed specifically at the decision uncertainty level preceding the action.

### Motor memory tagged by decision uncertainty contexts
A more direct test of context-dependent motor learning is to show that participants can differentiate between two different force fields associated with different contexts for the same reaching movement[8,18]. In Experiments 2 and 3, which used a within-participant design, we directly examined whether decision uncertainty could function as a contextual cue to tag different motor memories, thus enabling the separation of the two different force fields.

Participants made decisions regarding motions with 100% and 3.2% coherence. After the baseline phase, the participants were exposed to two different force fields: strong and weak (different magnitudes) for Experiment 2-1 ($n = 19$; Fig. 2a) and two opposing (conflicting) force fields (clockwise (CW) or counterclockwise (CCW)) for Experiment 3-1 ($n = 18$) and 3-2 ($n = 14$) (Fig. 3a). In both experiments, two different decision uncertainties were associated with either of the two different force fields.

Supposing that the brain can use decision uncertainty to separate contexts and assign relevant motor memory to each context. Then for force fields of different magnitudes (Experiment 2-1), participants should be able to produce different 'magnitudes' of resisting force, depending on the uncertainty context. Similarly, for opposing force fields (Experiment 3-1, 3-2), the participants should be able to produce an opposing resisting force in the direction (CW or CCW) associated with the given uncertainty context.

If uncertainty could not be used as a contextual cue for the different-magnitude force fields, participants would produce the same amount of resisting force, regardless of the decision uncertainty level. For opposing force fields, owing to the interference between opposing force directions, no systematic force compensation should be observed in either decision uncertainty context.

For force fields with different magnitudes (Experiment 2-1), after learning, participants successfully differentiated the exerting force magnitude according to the given decision context, producing a stronger force for the strong-force condition (mean±s.d.: $51.1 \pm 11.7\%$) than for the weak-force condition ($40.0 \pm 8.0\%$) (paired $t$-test, $t_{(18)} = 7.40$, $P < 0.001$, dz = 1.70) (Fig. 2c,d; see also Supplementary Fig. S4a). If the visual input pattern itself (100% and 3.2% coherent motion), and not decision uncertainty, is the tag for motor memory, merely associating these visual stimuli with different force fields should also lead to the same result. However, when random-dot motion with different coherence levels was associated with different force fields, but without involving any motion direction decisions (Experiment 2-2, Fig. 2b; see Methods), the difference became significantly weaker compared with the effect found in Experiment 2-1 (two-sample $t$-test, $t_{(37)} = 4.27$, $P < 0.001$, dz = 1.13) (Fig. 2e; see details in Supplementary Fig. S2c,d). This indicates that the results of Experiment 2-1 cannot be simply explained by the different patterns of visual input, corroborating previous findings[18]. We further confirmed that the difference in stimulus duration could not fully explain the results (two-sample $t$-test, $t_{(34)} = 4.26$, $P < 0.001$, dz = 1.17) (Experiment 2-3, Fig. 2e and Supplementary Fig. S2e,f, see Methods). There was also no systematic relationship between the difference in reaction time (random-dot motion onset to movement onset) and context separation level (difference in force output level between conditions) in Experiment 2-1 ($R = 0.035$, $P = 0.886$), supporting the minor role of stimulus duration in the current contextual effect.

We conceptually replicated the results of Experiment 2 (dual-magnitude force fields) for opposing force fields (Experiments 3-1 and 3-2). Participants successfully produced force in opposing directions depending on the uncertainty context in Experiment 3-1 (certain-decision: $21.7 \pm 21.2\%$ (s.d.), uncertain-decision: $-16.1 \pm 30.4\%$, paired $t$-test, $t_{(17)} = 5.91$, $P < 0.001$, dz = 1.4) (Supplementary Fig. S2k,l), as well as in Experiment 3-2, where the strength of the force field was doubled compared with Experiment 3-1 (certain-decision: $18.2 \pm 20.2\%$, uncertain-decision: $-9.3 \pm 28.0\%$, paired $t$-test, $t_{(17)} = 5.05$, $P < 0.001$, dz = 1.26) (Fig. 3c,d; see also Supplementary Fig. S4e). Furthermore, in the control condition (Experiment 3-3), when the visual feature (100% or 3.2% coherent motion) was associated with the force-field direction but without involving any decision (Experiment 3-3, Fig. 3b and Supplementary Fig. S2i,j), the compensation level was significantly weaker than in the main experiments (two-sample $t$-test, Experiment 3-1 vs 3-3; $t_{(31)} = 4.91$, $P < 0.001$, dz = 1.31; Experiment 3-2 vs 3-3; $t_{(29)} = 4.28$, $P < 0.001$, dz = 1.23) (Fig. 3e).

Finally, there was no significant difference between the peak velocity of the movements between different force magnitude conditions (Experiment 2-1: paired $t$-test, $t_{(18)} = 1.73$, $P = 0.100$, dz = 0.40) or between the two opposing force field conditions (Experiment 3-1: paired $t$-test, $t_{(17)} = 1.23$, $P = 0.237$, dz = 0.28; Experiment 3-2: paired $t$-test, $t_{(15)} = 0.13$, $P = 0.897$, dz = 0.03), indicating that movement velocity is not the key factor for the contextual separation. Taken together, these results show that preceding decision uncertainty works as a cue to contextualize and retrieve distinct motor memories.

### The context of decision uncertainty is stimulus-invariant
We further investigated what constitutes this type of novel uncertainty context, whether it is tied to the uncertainty of a specific input stimulus (for example, random-dot motion), or whether it is a stimulus-invariant, abstract uncertainty about the decision. In the latter case, participants should be able to retrieve the motor memory if the decision uncertainty level matches the learning and test phases regardless of the mismatch of the visual stimulus.

To examine this, we used two types of visual stimuli in Experiment 4: one was a random-dot motion, as was used in previous experiments

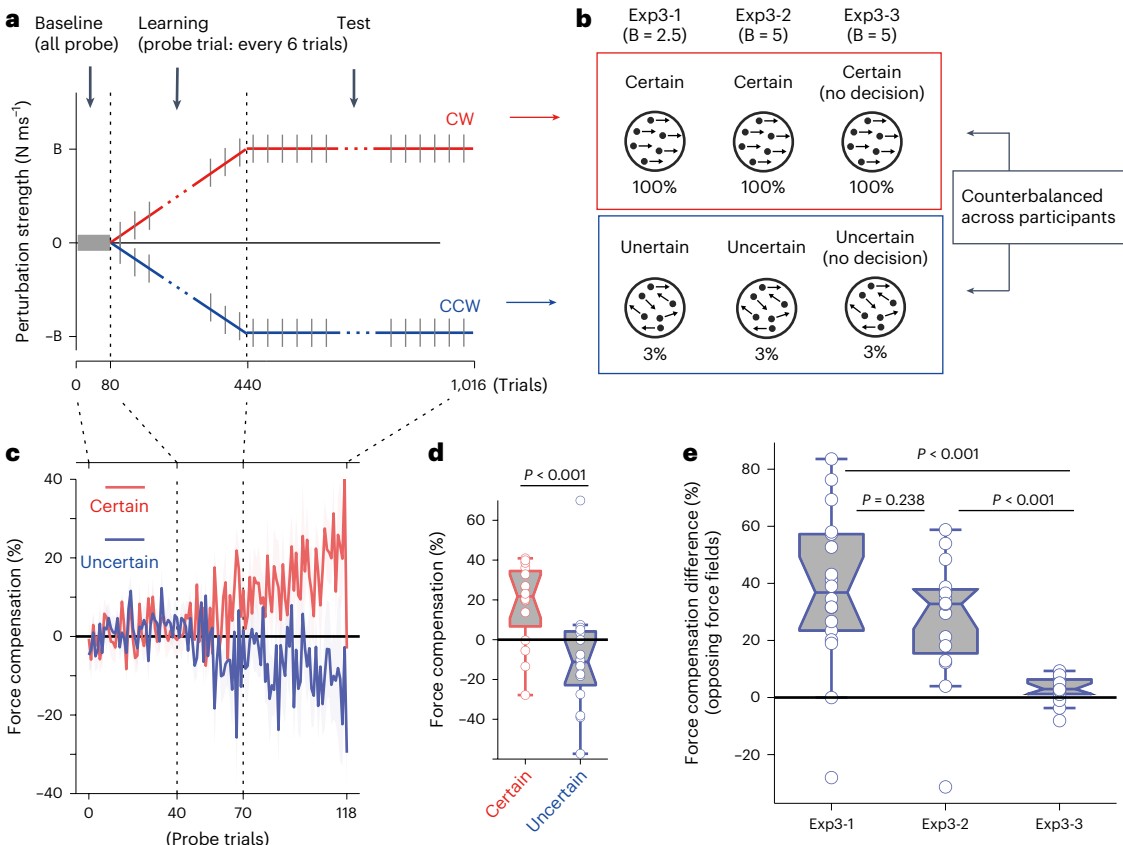

**Fig. 3 | Simultaneous learning of opposing force fields based on the decision uncertainty context (Exp.3).** Results of force field learning for two opposing force fields. **a**, Structure of the experimental trial. Participants simultaneously learned two different force fields in opposite directions (CW and CCW), each associated with different decision uncertainty levels. The grey line indicates probe error-clamp trials. **b**, Conditions of the main (Exp. 3-1, 3-2) and control (Exp. 3-3) experiments, where B indicates the N ms⁻¹ value of the force field. In Exp. 3-1 ($n$ = 18), certain (100% coherent motion) and uncertain (3% coherent motion) decisions were associated with two opposing force fields. The strength of the force field doubled in Exp. 3-2 (B = 5) ($n$ = 16) compared with Exp. 3-1 (B = 2.5). In Exp. 3-3 ($n$ = 15), 100% and 3% motion stimuli were each associated with different force fields; however, the participants did not make any motion direction decisions and only reached a specified target after the stimulus

disappeared (B = 5). The association between uncertainty level and force-field strength was counterbalanced across participants. **c,d**, Force compensation level measured in probe error-clamp trials in Experiment 3-2 (two-sided paired $t$-test). The data were aligned to motion coherence levels (100% and 3.2%). In Experiment 3-2, the force-field strength gradually increased during the learning phase. **e**, Comparison of the effect with control experiments, expressed as the difference in the force compensation level between the two force fields (two-sided two-sample $t$-test, corrected). Error bars indicate the s.e.m. across participants. In the boxplots, each dot represents a participant, the midline of the box represents the median, the box limits span from the 25th to the 75th percentile and the whiskers indicate the range (minimum to maximum) of the data. Outliers are determined by data points that are greater than 1.5× size from the box.

(motion stimulus), while the second was an arrow stimulus in which a sequence comprising left and right arrows was presented in the centre of a screen for a short period of time (20 arrows in 1,500 ms) (Fig. 4a). In the arrow stimulus, participants were asked to decide which of the two stimuli (left or right arrow) was presented more frequently after the termination of the sequence and then immediately reach the target in the direction of their arrow decision. The uncertainty was manipulated by changing the ratio of the left and right arrows in the sequence. Before the main experiment, we matched the confidence level of decisions (the subjective estimate of decision uncertainty) between the random-dot motion and arrow stimuli on the basis of the participants' confidence reports from a separate experiment. The decision confidence of the arrow stimulus with a left:right ratio of 5.5:4.5 (5% bias from chance) corresponded to a 3% coherence-level random-dot motion stimulus. Likewise, a ratio of 9:1 (40% bias from chance) in the arrow stimulus corresponded to 100% coherence-level random-dot motion (Fig. 4c).

As in Experiment 2-1, the participants learned two different magnitudes of force fields (strong and weak) associated with motion stimuli with either 100% or 3% coherent motion. After learning, we tested whether the arrow sequence stimuli could retrieve the force learned under the random-dot motion stimuli with matched confidence

levels. Importantly, during the task, all arrow-stimulus trials were error-clamped probe trials (grey lines in Fig. 4b). This indicates that the participants never experienced force perturbations while performing the action following the arrow stimulus decision. Therefore, any force produced to resist perturbation during the arrow stimulus is a component transferred from the motor memory formed by the random-dot motion stimulus.

First, we replicated the results of Experiment 2-1. The participants were able to learn two different force fields associated with two different uncertainty levels in the motion decision. The amount of force produced during the probe trials was significantly different between the two different force-field conditions (strong mean ± s.d.: 51.3 ± 18.6%; weak: 42.9 ± 12.5%) (paired $t$-test, $t_{(17)}$ = 4.47, $P$ < 0.001, dz = 1.05) (Fig. 4d left and Supplementary Fig. S3a). Second, and more critically, for trials with arrow decisions, we further found a significant difference in the force between the strong and weak force field conditions (strong: 46.5 ± 15.5%; weak: 42.9 ± 13.6%) (paired $t$-test, $t_{(17)}$ = 3.47, $P$ = 0.003, dz = 0.81) (Fig. 4d right and Supplementary Fig. S3b). Finally, the individual differences in force compensation between the two force fields (the index of contextual separation) were correlated between the random-dot motion and arrow sequence conditions (Fig. 4e; $R_{(18)}$ = 0.48;

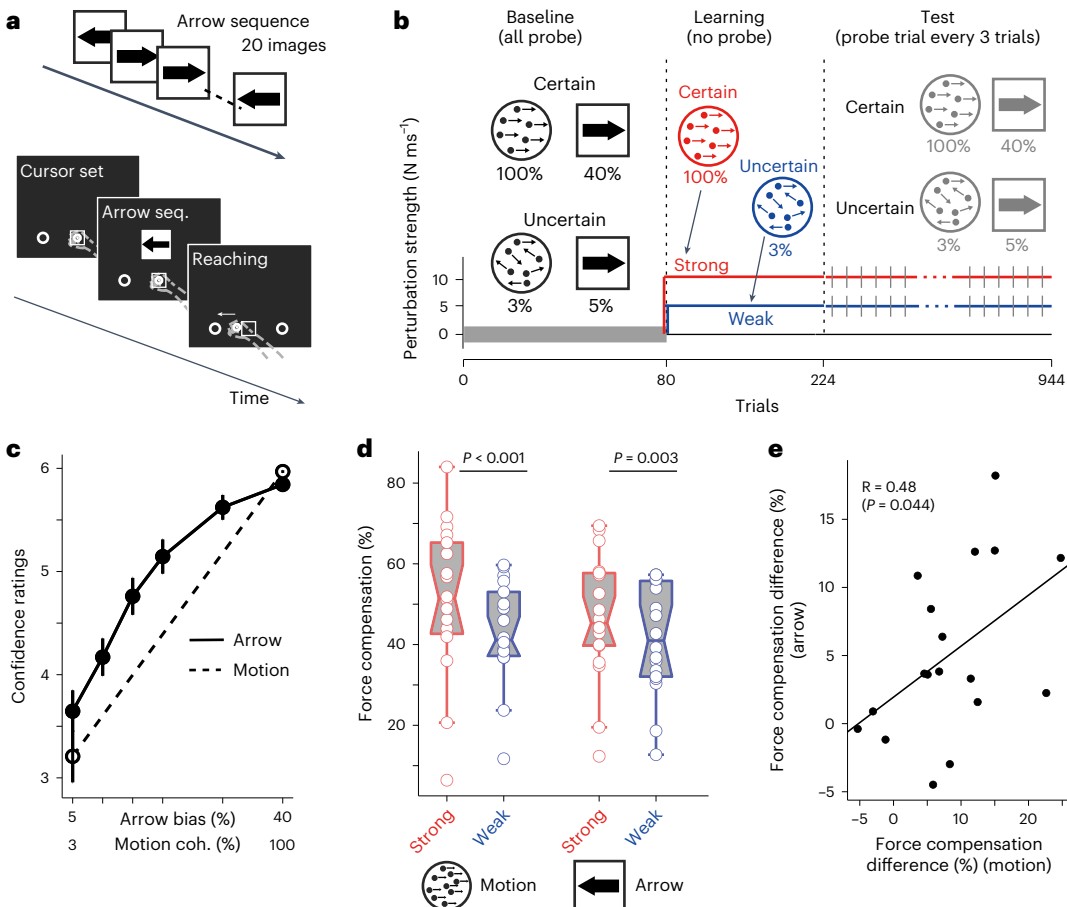

**Fig. 4 | Transfer of decision uncertainty context across different visual stimuli (Exp. 4). a**, Arrow sequence stimulus in Experiment 4 (*n* = 18). A sequence of 20 arrows was presented at the centre of the screen, after which the participants judged which arrow direction (left or right) had the highest probability in the sequence. Participants reached the target direction that matched their arrow direction decisions. **b**, Conditions and trial structure for Experiment 4. The grey line indicates probe error-clamp trials. In the baseline phase, participants made motion or arrow direction decisions and reached the target without perturbation. For both visual stimuli, certain and uncertain stimuli were prepared, in which the confidence level was matched across the two types of visual stimuli (**c**). In the learning phase, the participants learned the force field only during the motion direction decision, in which strong and weak force fields were associated with different decision uncertainty levels (certain and uncertain). In the error-clamp trials of the test phase, together with certain and uncertain motions, certain and uncertain arrow sequence stimuli were

presented again as in the baseline phase. Note that the reaching following the arrow decision in the test phase was performed only for the probe error-clamp trials (grey lines). The association between uncertainty level and force-field strength was counterbalanced across participants. **c**, Matching the confidence level across different types of visual stimuli (random dot and arrow sequence). **d**,**e**, Force compensation for each condition (two-sided paired *t*-test). The force field tagged with different motion uncertainty levels (left) was retrieved using different visual stimuli (arrow stimulus; right) with matched uncertainty levels (**d**). Across participants, the difference in the force compensation level (that is, the index of contextual learning) in the arrow stimulus correlated with that in the motion stimulus (**e**). Error bars indicate the s.e.m. across participants. In the boxplots, each dot represents a participant, the midline of the box represents the median, the box limits span from the 25th to the 75th percentile and the whiskers show the range (minimum to maximum) of the data. Outliers are determined by data points that are greater than 1.5× size from the box.

C.I. = 0.02, 0.77; *P* = 0.044), suggesting a shared component between the two variables. These results clearly demonstrate that the motor memory encoded by random-dot motion can be retrieved using different visual stimuli with similar decision uncertainty levels. In other words, part of motor memory is tied to abstract decision uncertainty, which is invariant to the features of the input stimulus.

## Decision uncertainty tags motor memory during planning

During perceptual decision-making, neural activity in the sensorimotor cortical areas reflects the evidence-accumulation process before action initiation[22,23]. Thus, the dynamics of neural activity patterns before an action differ when the decision is certain or uncertain[24]. If neural activity during the deliberation period of a decision (planning of action), which reflects the different uncertainty levels of the decision, is the index for uncertainty-based motor memory, eliciting the relevant activity pattern before the decision should be sufficient to

retrieve the associated motor memory. In Experiment 5, we examined this hypothesis by adopting the procedures of 'follow-through' experiments[6,7] to separate the planning and execution of actions.

The participants judged the direction of random-dot motion (left or right) (Fig. 5a). They made their left–right decision within 1,000 ms of the stimulus being presented. After the stimulus disappeared, they first made a straight reach to the central target and then made a follow-through movement to the left or right target according to their decision. In this design, during the first reaching of the central target, decision-related movement (left or right follow-through) is yet to be executed, and only the plan of the movement exists. Force-field perturbations were applied only during the first reaching movements. Two different decision uncertainty levels (certain: 100% coherent motion; uncertain: 3% coherent motion) were associated with two opposing force fields (CW or CCW) (Fig. 5b). Note that the different force fields are not associated with the follow-through 'directions', as has been done

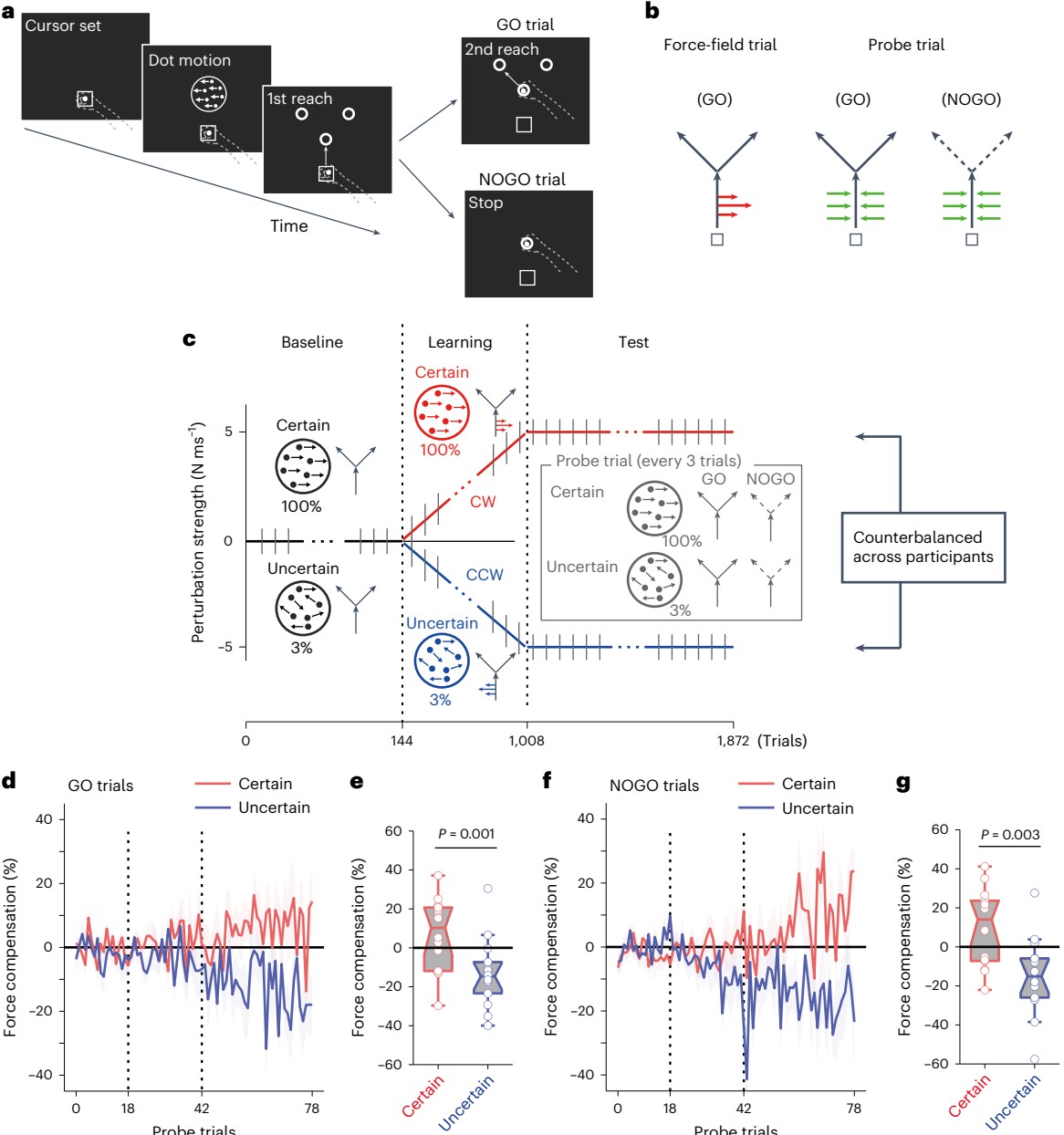

**Fig. 5 | Decision uncertainty context during the deliberation/planning period of decision making (Exp.5). a–c**, Schematic of the task structure in Experiment 5 (*n* = 14). The participants performed a follow-through reaching task after making a motion direction decision. The 1st reach was common regardless of the decision direction, but the 2nd reach (follow-through) was made in accordance with the participants' decisions (GO trials). Occasionally, the participants were instructed to stop at the 1st target during the movement (NOGO trials) (**a**). For the GO trials, opposing force fields were set on the 1st reaching path, depending on the decision uncertainty level, with occasional probe trials (**b**). The NOGO trials included only the probe trials. The strength of the force field gradually increased during the learning phase (**c**). **d–g**, Force compensation in probe error-clamp trials for both the GO trials (**d,e**) and NOGO trials (**f,g**) (two-sided paired *t*-test). Error bars indicate the s.e.m. across participants. In the boxplots, each dot represents a participant, the midline of the box represents the median, the box limits span from the 25th to the 75th percentile and the whiskers show the range (minimum to maximum) of the data. Outliers are determined by data points that are greater than 1.5× size from the box.

in previous studies[6,7]; here, even when making a decision in the same direction, the first reaching can be perturbed differently, depending on the uncertainty of the decision.

To further dissociate the plan and execution of the follow-through movement, in some trials after the participants initiated their actions, they were cued to stop at the first target (performing only the first reaching movement); these were classified as NOGO trials. NOGO trials were conducted to confirm that uncertainty-based motor memory can be retrieved with the plan of a (certain or uncertain) decision-relevant movement, even without fully executing it. Therefore, participants were never exposed to force fields during the NOGO trials, and all NOGO trials were error-clamp probe trials (Fig. 5b,c).

If the difference in the quality of the plan based on certain or uncertain decisions is sufficient to tag the motor memory, participants should be able to learn the opposing force fields, even when applied before the decision-relevant movement. Similarly, such learning should be transferred to NOGO trials, in which the decision is never overtly expressed and the participants have never experienced the force field.

As expected, the participants were able to separate the context for the two opposing force fields (paired *t*-test, $t_{(13)}$ = 4.12, *P* = 0.001,

dz = 1.10) (Fig. 5d,e). More importantly, the same motor memory was transferred and retrieved in the NOGO trials (paired *t*-test, $t_{(13)}$ = 3.66, *P* = 0.003, dz = 0.98) (Fig. 5f,g). This result cannot be explained by the participants using the target direction as the contextual cue[7], as the force field was not associated with the target direction, but rather with the uncertainty level of the stimuli. Here, even the same follow-through movement could be perturbed by different force fields; however, the participants were able to separate the motor memory depending on decision uncertainty.

Overall, the results of Experiment 5 support our claim that the neuronal activity pattern during the deliberation/planning phase, which reflects decision uncertainty, is likely to be a tag for context-dependent motor learning based on decision uncertainty.

## Discussion

The context for memory encoding is of great interest in the field of cognitive neuroscience[13,25–27]. In the domain of motor memory, the majority of the identified contexts are directly involved in the overt or ongoing motor control process, such as the spatial position of the workspace[18], direction of the planned movement in the workspace[5], plan of the future state[6,7] or concurrent state of the relevant or irrelevant body parts[8–10]. Our study further demonstrated that covert internal decision processes, without any overt difference in bodily state, could also be a important contextual cue for motor memory.

The current theory of context-based motor learning, including the recent COIN model[27], postulates that learners infer context on the basis of observed sensory cues and state feedback. According to this framework, context uncertainty leads to interference between associated motor memories. The present study challenges this notion by demonstrating that the uncertainty inherent in the decision process can serve as a context for motor memory. We further showed that the features of the visual stimulus, duration of the visual stimulus and movement velocity of the action could not explain our results. More importantly, in Experiment 4, we revealed that decision uncertainty itself is a contextual cue, rather than an uncertainty tied to a specific sensory input. A previous study showed that differences in action plans, and consequently, the underlying neural states, could function as a contextual cue[7]. Our study further illustrates that it is not only sensory inputs that directly specify future action plans, but that the inferred quality, or uncertainty, of the decision/motor plan can also play a role in contextualizing motor memory. In summary, our findings expand our understanding of how contextual inference influences motor memory, broadening the scope of contextual cues from physical input to include internal inferential processes, such as decision uncertainty.

Although we can quantify the overall effect of uncertainty by analysing choice patterns, the trial-by-trial subjective uncertainty levels may vary. For example, when deciding the motion direction for a 3% coherent motion stimulus, confidence may vary across trials, sometimes higher and sometimes lower. Therefore, 'uncertainty' appears to be an unstable tag for motor memory. Despite this, we consistently observed a statistically strong effect (dz > 1.0) of the decision uncertainty context across multiple experiments throughout this study. Furthermore, the average compensation level of the force field was comparable to that reported in a previous study[6], particularly when a challenging combination of contexts (follow-in and follow-through) was associated with different force-field patterns (compensation level at the end of Day 1; <20%). Taken together, these results indicate the robustness of the decision uncertainty context to motor memory.

During perceptual decision-making, the ongoing accumulation of evidence during the deliberation period is reflected in neural activity in the cortical areas involved in motor planning and execution[22,23,28–30]. Perceptual evidence guides an agent's decision, but the agent can simultaneously calculate the subjective uncertainty level of the decision (that is, decision confidence) using accumulated evidence signals[31]. Neural activity in both the cortical[31] and subcortical structures reflects

the uncertainty of the action to be performed[32,33]. As such, it is possible that this pre-movement neural activity pattern, which reflects decision uncertainty, enables the brain to form different motor memories within the sensorimotor network. Experiment 5 provides further support for this hypothesis by demonstrating that motor memory formed in a particular uncertainty context can be retrieved when performing the same decision, without fully executing it. This is concordant with the claim that the neural state in the sensorimotor areas indexes motor memory[34,35] and with the cue-based switching of internal models in the motor areas[36]. Taken together, the neural state during decision making not only reflects the internal trajectory towards the final decision but also specifies the action to be retrieved during that decision state.

Previous studies have shown that actions learned while concurrently performing an irrelevant attention-demanding task can be retrieved better during the same dual-task situation[37,38], indicating the existence of a context-dependent attentional filtering mechanism for motor learning. In the present study, the outcome of decision making was directly relevant to the subsequent action; therefore, the underlying computation may be fundamentally different from that in the previous dual-task scenario. However, both studies demonstrate that the brain optimizes the motor control strategy on the basis of current cognitive demands. Future studies are required to classify the family of cognitive computations that enables the formation of the sensorimotor context, together with the underlying neural mechanisms.

Uncertainty regarding how an action will be perturbed affects motor learning, where the learning rate is modulated depending on the stability of the environment[39]. This phenomenon cannot explain our results because the amount of learning itself did not depend on the uncertainty level of the decision (Fig. 1e). This indicates that coping with the uncertainty of decisions and the uncertainty of perturbations are both governed by different processes in the brain; for the former, the brain contextualizes motor memory depending on decision uncertainty.

Recent studies have highlighted the existence of two motor learning systems: an explicit learning system (that is, explicitly changing the aim of control) and an implicit system (that is, recalibrating the parameters of motor control)[40,41]. It is unclear whether motor memory formed on the basis of the decision-uncertainty context is acquired through an implicit or explicit learning process. Although a recent study using visuomotor rotation task showed that contextual motor learning can be achieved using an explicit learning strategy[42], we believe that the contextual learning based on decision uncertainty in this study involves an implicit learning system for two reasons. First, we used force-field adaptation for the task, where the effect of explicit strategy on learning has been shown to be limited compared with visuomotor adaptation tasks[43]. Second, in the two experiments (Experiments 2 and 4), the participants were able to separate strong and weak force fields according to the decision uncertainty context. Because the two perturbations are both in the same direction, it would be very demanding to change the magnitude of the resisting force explicitly or consciously, depending on the preceding uncertainty of the decision. Therefore, we believe that the major contributor to learning in the present study was the implicit component of learning. However, future studies are required to clarify which type of contextual motor learning is more prone to explicit strategies.

We have demonstrated that decision uncertainty can function as a contextual cue for motor memory; however, the underlying neural computations and brain regions that enable such indexing remain to be elucidated. To overcome this limitation, the necessary next step would be to integrate the present task with neural recordings.

In conclusion, we showed that the brain uses decision uncertainty as a contextual cue to retrieve motor memories, thus preparing different motor memories depending on the uncertainty level of decisions. This indicates that football players should practice kicking the ball precisely to the place they want in both situations when they are sure and unsure about the goalkeeper's movement.

## Methods

### Participants

A total of 189 right-handed participants volunteered in Experiment 1 (certain group: 22 (7 women), ages 19–25 years; uncertain group: 22 (7 women), age 20–25 years), Experiment 2-1 (21 (5 women), age 20–28 years), Experiment 2-2 (20 (7 women), age 20–38 years), Experiment 2-3 (17 (5 women), age 21–29 years), Experiment 3-1 (20 (8 women), age 20–30 years), Experiment 3-2 (16 (5 women), age 21–38 years), Experiment 3-3 (15 (3 women), age 20–30 years), Experiment 4 (20 (7 women), age 21–46 years) and Experiment 5 (16 (5 women), age 20–25 years). All participants were naïve to the purpose of the experiment. All experiments were performed with the understanding and written consent of each participant, in accordance with the Code of Ethics of the World Medical Association (Declaration of Helsinki). All experiments were conducted at the Center for Information and Neural Networks (CiNet), National Institute for Information and Communications Technology (NICT), with all protocols approved by the NICT ethics committee (N230062305). No adverse events occurred in any of the experiments. Experiment 1 used a relatively larger sample size for each group compared with typical motor learning studies because of the cross-participant design. To ensure an effect size similar to that in Experiment 1, we used a similar number of participants for the remaining experiments. Participants were compensated ¥1,000 per hour for their participation in the experiments.

### Data and participant exclusion criteria

In each experiment, trials were excluded if (1) reaction times (movement onset concerning the visual stimulus onset) were too fast (<100 ms; likely not judging the stimulus) or too slow (>1,500 ms; judging after the stimulus disappearance), (2) movement did not reach the target (<75% of the maximum distance), or when the movement direction reversed after moving 2.5 cm in the opposite direction before reaching the target. If the trial exclusion rate exceeded 30% of the data in the last block of the learning phase or test phase, the participants were excluded from further analysis. Furthermore, if the overall choice rate during the test phase was biased towards one direction (>70%) (for example, moving (making a decision) to the right in most trials), the participant was excluded because of an asymmetrical motor learning experience between the two directions (see Methods section below for task details). These exclusion criteria were set to exclude data/participants who did not follow the instructions of the experiments and to maintain the same data quality across participants. However, the exclusion of participants from the analysis did not qualitatively affect the results.

On the basis of the above criteria, three participants from each certain and uncertain group were excluded from Experiment 1. Similarly, two participants each from Experiments 2-1, 3-1, 4 and 5 were excluded.

### General settings

The participants were seated comfortably in front of a screen placed horizontally in front of them, which prevented direct visualization of their hands (Fig. 1a). A visual stimulus was presented on a screen using a projector placed above the screen. The viewing distance was set at 50 cm. The upper trunk was constrained using a harness attached to a chair to maintain the viewing distance. During the experiment, participants were asked to hold the handle of the manipulandum with their right hand (PHANToM Premium 1.5 HF; SensAble Technologies), whose position was sampled at 500 Hz. The handle position was displayed as a white cursor (circular, 6 mm in diameter) on a black background on a horizontal screen located above the hand. The movement of the handle was constrained to a virtual horizontal plane (10 cm below the screen), which was implemented using a simulated spring (1.0 kN m$^{-1}$) and dumper (0.1 N ms$^{-1}$).

A random-dot motion stimulus was presented at the centre of the screen[44,45] (Fig. 1a). In a circular aperture with a diameter of 7°, the dots were presented at a density of 3.5 dots deg$^{-2}$. The speed of the dots was 10° s$^{-1}$. Each dot appeared at a random position inside the aperture, moved in a specific direction for 133 ms and then disappeared (lifetime). The disappeared dot reappeared at a random position in the aperture and moved in a reassignment direction. For each trial, 3.2%, 6.4%, 12.8%, 25.6%, 51.2% and 100% of the dots moved coherently to the left or right, respectively (hereafter referred to as the motion coherence level). All other dots moved in random directions and were selected for each dot separately between 0° and 360°. Control of the robotic manipulandum (haptic device; PHANToM Premium 1.5 HF) and the associated visual stimuli were programmed using C++ (Visual studio v.2008)[5].

Before each trial, the robotic manipulandum automatically guided the participant's hands to the starting position. The trial began when the participants maintained the cursor at the starting position for 500 ms. Subsequently, a random-dot motion was displayed. Immediately after the decision, the participants made a reaching movement towards either the left or right target, depending on their decision (Experiments 1, 2 and 3). Movement onset was defined as the point at which the tangential velocity of the movement reached 10% of the peak velocity. The motion stimulus disappeared when movement was initiated. In Experiments 4 and 5, participants were required to move after the disappearance of the motion stimulus. Each target was located 10 cm horizontally to the starting position.

A velocity-dependent curl force field[14] was used for motor learning. A force field was applied according to the following equation:

$$\begin{bmatrix} f_x \\ f_y \end{bmatrix} = \begin{bmatrix} 0 & B \\ -B & 0 \end{bmatrix} \begin{bmatrix} v_x \\ v_y \end{bmatrix}, \tag{1}$$

where $f_x$ and $f_y$ are the forces applied to the handle (N), and $v_x$ and $v_y$ are the velocities of the handle (m s$^{-1}$) in the $x$ and $y$ directions, respectively. For the clockwise force field, the viscosity coefficient $B$ (N ms$^{-1}$) was positive, whereas for the counterclockwise field, $B$ was negative. Channel trials (error-clamp probe trials) were occasionally introduced to quantify the learning of the force field. During the channel trials, the handle movement was constrained along a straight path between the home position and the target by a simulated damper and spring[16], and the force applied to the wall of the channel was measured. This allowed us to measure the force retrieved to resist perturbations in a given context while avoiding kinematic errors.

### Experiment 1

We tested how the action learned under a particular level of decision uncertainty was transferred to actions during other levels of decision uncertainty.

**Procedure.** Participants held the handle with their right hand and judged the direction of the random-dot motion (left or right). As soon as they made the decision, the participants moved their hands towards the target direction corresponding to the direction of judgement. The random-dot motion disappeared immediately after the participant's movement was detected (3.5 cm s$^{-1}$). The stimulus disappeared after 1,500 ms, even if no movement was detected[45], and the participants were instructed to initiate their movement before the disappearance. Before the task, the participants were familiarized with the manipulandum and the judgement of the visual stimulus.

The experiment comprised two phases: learning and test. In both phases, the task was performed under a force field, with occasional error-clamp trials (Fig. 1c). Half of the participants experienced the CW force field and the other half experienced the CCW force field. The viscosity coefficient ($B$ in equation (1)) was set to 10 (N ms$^{-1}$).

Participants were divided into two groups: certain and uncertain. During the learning phase, the participants in the certain group learned the force-field reaching following a 100% coherent motion decision (low decision-uncertainty level). Participants in the uncertainty group

learned to reach following a 3.2% coherent motion decision (high uncertainty level). Participants were instructed to maintain a straight movement trajectory similar to that of reaching without perturbation. Five blocks of 72 trials each were conducted. Error-clamp trials were introduced every six trials between the force-field trials. The motion coherence level during error-clamp trials was set to be the same as that during force-field trials.

In the test phase, each group performed the same task as in the learning phase. The only difference was that the frequency of error-clamp trials was, on average, every three trials, while six different coherence levels (±3.2%, 6.4%, 12.8%, 25.6%, 51.2% and 100%) were used (positive values indicate direction towards the right and negative to the left). This design allowed us to examine how motor memory formed at a particular decision uncertainty level was generalized to other levels of uncertainty. Participants underwent 9 blocks, with each block containing 66 trials (22 error-clamp trials; 2 (left and right) trials for 100% coherent motion, 2 trials each for the other 10 motion coherence levels, and 44 force-field trials). Each block took around 7 min to complete. After each block, the participants were asked whether they needed any breaks. If requested, they were allowed to take breaks (maximum of 2 min) between the blocks without leaving the seat. No special break was prepared in between the learning and test phases. Overall, the duration of the experiment was around 135 min, including the practice session.

It has been shown that even in error-clamp trials, individuals gradually forget force-field learning[16]. Therefore, we varied the uncertainty levels of the probe trials during the test phase, but the participants continued to experience the force-field trials with the assigned decision uncertainty levels (100% or 3.2%) to maintain the learning of the decision-uncertainty-tagged motor memory.

### Experiment 2-1
To directly demonstrate the role of decision uncertainty as a contextual cue for motor memory, we tested whether the participants could learn two different force fields for the same movement trajectory if each force field was associated with different decision uncertainty levels.

**Procedure.** As in Experiment 1, the participants judged the direction of a random-dot motion and moved the handle towards the target in the judged direction. Two motion coherence levels were prepared: 100% (certain decisions) and 3.2% (uncertain decisions). In the baseline phase, after the practice session, participants performed an error-clamped task (2 blocks of 40 trials). In the learning phase, participants performed the task under two different force-field strengths ($B = 10$ (strong) and $B = 5$ (weak) (N ms$^{-1}$)). Each strong and weak force field was associated with a different preceding decision uncertainty (certain or uncertain). The pattern of association between force-field strengths, decision uncertainties and the direction of the force fields (CW or CCW) was counterbalanced across participants. Participants underwent 2 blocks of 72 trials each. In the test phase, the participants performed 10 blocks of the task (72 trials) with interleaved error-clamp trials (6 trials each for two visual stimuli per block).

### Experiment 2-2
As a control experiment, we examined the contextual effects of visual features (100% and 3.2% coherent random-dot motion), which correlated with decision uncertainty in Experiment 2-1.

**Procedure.** The setting of the experiment was the same as in Experiment 2-1, but the participants were not required to make any directional decisions regarding the random-dot motion. Instead, they saw either 100% or 3.2% coherent random-dot motion presented on the screen. Immediately after the disappearance of the motion stimulus, a single target appeared on either the left or right side and the participants reached the target. The target direction did not correlate with the direction of the participants' movement

and decision were unrelated, which discouraged them from making decisions in any direction. The duration of the visual stimulus was sampled from a normal distribution, where the mean and variance were extracted from the reaction times (RT: stimulus onset to movement onset) in Experiment 2-1 (Supplementary Fig. 2b; parameters: 3.2% motion: 461.9 ± 75.8 ms (left), 453.9 ± 69.9.2 ms (right); 100% motion: 731.5 ± 144.3 ms (left), 720.3 ± 149.2 ms (right)). To ensure that the participants did not ignore the stimulus, they were occasionally asked whether the visual motion they saw was coherent or random (12 trials per block). The average correct rate for this task was 91.4 ± 12.7%.

All the other trial structures were identical to those in Experiment 2-1. After the baseline condition (2 blocks of 40 trials; all error-clamped), in the learning and test phases, each coherence level of the random-dot motion was associated with either a strong or weak force field in each participant (learning phase: 2 blocks of 72 trials; retrieval phase: 10 blocks of 72 trials (1 error-clamp every 6 trials)).

A comparable level of force-field learning as in Experiment 2-1 should be observed if the visual feature of the stimulus itself can be a context for the encoding/retrieval of motor memory.

### Experiment 2-3
In another control experiment, we examined the contextual effect of time before execution, which also correlated with the decision uncertainty level in Experiment 2-1.

**Procedure.** The setting of the experiment was the same as in Experiment 2-1, but the participants observed only 100% coherent random-dot motion. Two durations were prepared, one of which corresponded to the RTs (stimulus onset to movement onset) of 100% coherent motion (short duration) and the other to the RTs of 3.2% coherent motion (long duration) in Experiment 2-1. As in Experiment 2-3, this duration was drawn from a normal distribution in which the mean and variance were extracted from the RTs of the corresponding conditions in Experiment 2-1 (see above).

In this experiment, the participants judged the direction of the visual stimulus and reached the target. The remaining parameters were similar to those in Experiment 2-1. After the baseline phase (2 blocks of 40 trials; error-clamped), in learning (2 blocks of 72 trials) and test (10 blocks of 72 trials; error-clamp, once in 6 trials) phases, short and long durations were associated with either weak or strong force fields, respectively.

Unlike Experiment 2-2 in which participants were uninformed of the movement direction until the disappearance of the random-dot motion, this experiment allowed participants to prepare the movement for a longer duration when the stimulus duration was longer. If the stimulus duration and the amount of motor preparation were the main components of the context in Experiment 2-1, we should observe an effect comparable to that in Experiment 2-1.

### Experiment 3-1
We tested whether two force fields in opposite directions could be learned simultaneously if each field was associated with different decision uncertainty levels.

**Procedure.** The experimental settings were identical to those in Experiment 2-1, except that instead of using strong and weak force fields, we associated two force fields with opposing directions (CW and CCW) and different decision uncertainties (100% and 3.2%, respectively). Also, participants underwent 3 blocks of 72 trials during the learning phase. The viscosity level was set to ±2.5 (N ms$^{-1}$) for the CW and CCW conditions.

### Experiment 3-2
We tested whether an effect comparable to that in Experiment 3-1 could be obtained even if the strength of the force field was twice that

of Experiment 3-1. After the baseline of 80 trials (all probe trials; 2 blocks of 40 trials), in the learning phase, force-field strength increased towards the two opposing directions (CW or CCW), trial-by-trial for each condition (100% or 3.2% random-dot motion) until it reached ±5 N ms$^{-1}$ (360 trials; 5 blocks of 72 trials). This strength persisted throughout the test phase (576 trials, 8 blocks of 72 trials). Throughout the learning and test phases, error-clamp probes were introduced once every 6 trials. Probe trials in the second half of the test phase (the final 4 blocks) were used to compare the force compensation level between the conditions to allow the participants to experience sufficient trials of the maximum force-field strength.

### Experiment 3-3

A control experiment was conducted for the opposing force experiment. The procedure was largely similar to that in Experiment 3-2, but instead of making directional decisions about the random-dot motion, participants viewed the stimulus and after the disappearance, they reached the target direction that appeared to the left or right (same as in Experiment 2-2). Two different random-dot motion stimuli, 100% and 3.2% coherent motion stimulus (not the movement direction), were each associated with CW or CCW force field (±5 N ms$^{-1}$).

### Experiment 4

We further examined whether the motor memory associated with the decision uncertainty context was stimulus-dependent or could be transferred to other visual stimuli with matched decision uncertainty levels.

**Procedure.** Two types of visual stimuli were prepared: random-dot motion and an arrow sequence. For random-dot motion, participants judged the net direction (left or right) of the dot motion. The decision uncertainty was controlled by changing the coherence percentage of the dot motion direction. The arrow stimulus consisted of a stream of arrows heading either left or right (Fig. 4a). A total of 20 arrows were presented in the centre of the screen in a sequence, each presented for 33.3 ms, followed by 33.3 ms of blank screen. The participants judged the direction of the arrow that was more frequently presented in the sequence. The uncertainty of the decision was manipulated by changing the left:right ratio of the arrows in the sequence.

**Matching of subjective uncertainty level across the stimuli.** First, we established a correspondence in the subjective uncertainty level (confidence) between the two stimuli. In each trial, either the random-dot motion stimulus or the arrow stimulus was presented for 1,500 ms before disappearing. After the disappearance of the stimulus, the participants moved the manipulandum towards the target in the direction of their judgement, and no perturbation was applied to this movement. After moving their hand to the target, participants reported the confidence level of the decision on a scale of 0 to 6, with 0 corresponding to a total guess and 6 corresponding to maximum confidence in the decision. Participants performed 5 blocks of 64 trials each. Two motion coherence levels (100% and 3%) were used for the random-dot motion stimulus. For the arrow stimulus, the left:right ratios in the arrow sequence were 55%, 60%, 65%, 70%, 80% and 90% (5–40% bias). Each block contained 16 random-dot-motion stimuli and 48 arrow stimuli.

**Testing transfer of motor memory across different stimuli.** In the confidence-matching experiment, we found that the decision confidence for the 5%-biased arrow sequence corresponded to a confidence of 3% coherent random-dot motion. Similarly, a 40% biased arrow sequence corresponded to a 100% coherent random-dot motion. Using these four confidence-matched stimuli, we tested the transfer of uncertainty-tagged motor memory across different visual stimuli.

In the baseline phase, all four types of stimulus were presented and the participants underwent 2 blocks of 40 trials (all error-clamped) (Fig. 4b). Next, in the learning phase, only two coherence levels of random-dot motion (100% and 3%) were presented, each of which was associated with either strong or weak force fields, as in Experiment 2-1. Participants performed 2 blocks of 72 trials each. Finally, all four stimuli were presented during the test phase. Here, random-dot motion stimuli had both force- and error-clamp trials; however, only error-clamp probe trials were used for the arrow stimuli. This prevented any learning of force for the arrow stimulus trials, allowing us to purely evaluate the components transferred from learning using a random-dot stimulus. Participants underwent 10 blocks of 72 trials (error-clamp probe trials, once every 3 trials).

If the uncertainty context includes an abstract, stimulus-invariant component, the motor memory tagged by the decision uncertainty of random-dot motion should be retrieved when the arrow stimulus with a matched uncertainty level is presented.

### Experiment 5

We examined whether the decision uncertainty context was readily represented in the planning/deliberation stages before executing the decision.

Participants performed a double-step follow-through reaching movement after deciding the direction (left or right) of the random-dot motion (Fig. 5a). A random-dot motion stimulus was presented for 1,000 ms and the participants were instructed to decide the direction while the stimulus was presented. After the random-dot motion disappeared, the participants first made an 8 cm reaching movement from the starting position to the central target. They were required to briefly stay at the central target for at least 50 ms and then make a secondary reaching movement to either of the two targets, left or right, depending on their decision. Secondary targets were positioned at +45° or −45° relative to the line connecting the starting position and the central target, and 8 cm away from the central target. Therefore, the first reach was identical irrespective of the decision they made in their minds, and their decision was executed in the secondary reach.

As in the other experiments, two different decision uncertainty conditions were prepared: certain (100% coherent motion) and uncertain (3% coherent motion). After the baseline period, the two opposing force fields (CW and CCW) were associated with either of these two uncertainty conditions. Importantly, the force fields were applied only during the first reaching movement and not during the secondary reaching movement (Fig. 5b). Thus, we were able to examine whether the quality of planning for the decision (certain or uncertain) can be a contextual cue for motor memory, by examining the first reaching movement. Note that the quality of planning (decision uncertainty) is independent of the direction of planning (target direction).

To ensure that decision uncertainty during the planning and deliberation of a decision can be a contextual cue for motor memory, in some cases, participants were cued to stop at the first central target while making movements (NOGO trials). The NOGO trials were cued by eliminating the two secondary targets when the participants' movements passed the midpoint (4 cm from the starting point) of the first reach.

Participants underwent 3 blocks (1 block, 72 trials) for the baseline, 4 blocks of increasing force field and 6 blocks of ±5 N ms$^{-1}$ force-field trials (Fig. 5c). Throughout the experiment, an error-clamp probe trial was introduced every 3 trials, half of which were NOGO trials. We used probe trials (certain vs uncertain comparison for each GO and NOGO trial) of the final 5 blocks for force compensation level comparison.

### Data analysis

**Data analysis of Experiment 1.** All data analysis was carried out using Matlab v.2020b (Mathworks). To calculate the compensation level of the force perturbation, the endpoint force against the channel wall (lateral force) in the error-clamp trials was extracted. We subsequently regressed the force time series with the time series of movement

velocity and position[46]. Perfect compensation would be the velocity coefficient value becoming equal to the force-field constant *B* (see equation (1)); therefore, we defined the compensation level as the velocity coefficient divided by *B*. Force compensation was calculated for all trials of all experiments.

To test the generalization of learning, we divided the force compensation level during probe trials in the test phase by the average force compensation level of the final block of the learning phase (12 channel trials) (Fig. 1e). The force compensation values across different uncertainty levels without normalization are shown in Supplementary Fig. S1e.

To quantify the differences in the decision-making process between the certain and uncertain groups, we fitted a DDM model to the RT and choice data for each group. This model accumulated momentary sensory evidence over time to form a decision variable (DV) until it reached an upper or lower bound. The bound reached and timing determine the choice and decision times. Reaction time was modelled as the sum of the decision time and additional sensory and motor delays (non-decision time). The model had three free parameters: sensitivity, bound height and mean non-decision time. The sensitivity *k* determines the linear scaling of the mean momentary evidence in the model with signed stimulus strength. The bound height, *A*, determines the amount of evidence that must be accumulated to reach the upper ($+A$) or lower ($-A$) bound. The non-decision time is drawn from a Gaussian distribution, whose mean ($T_0$) is a free parameter, and the standard deviation is set to 30% of its mean.

We fit the DDM to the behavioural data of individual participants using a maximum-likelihood estimation. On the basis of the above formulation, the probability of crossing the upper and lower bounds can be derived numerically by solving the Fokker–Planck equation[47]. The resulting bound-crossing probability was convolved with the distribution of non-decision times to obtain the probability distributions of choices and RTs for each stimulus strength. These distributions were used to calculate the log-likelihoods of the observed choice and RT for each trial, which were then summed across trials to search for the best set of parameters that maximized the sum for each participant. The obtained model parameters were $k = 0.27 \pm 0.08$ (s.d.), $A = 12.4 \pm 3.2$, $T_0 = 483 \pm 100$ for the certain group, and $k = 0.43 \pm 0.21$, $A = 15.2 \pm 3.5$, $T_0 = 491 \pm 98$ for the uncertain group (Supplementary Fig. S1c,d). The model curves presented in Supplementary Fig. S1a,b were generated by averaging the fitting results for each participant.

**Data analysis of Experiments 2 and 3.** All forces measured during the error-clamp probe trials were transformed into force compensation levels (see the analysis of Experiment 1). In Experiments 2-1, 2-2 and 2-3, the coefficient was calculated on the basis of the force in the strong force condition to allow a direct comparison between the two force conditions. Thus, successful learning in the strong condition resulted in a compensation value of 100% and in the weak condition, a compensation level of 50%. For Experiments 3-1, 3-2 and 3-3, the compensation relative to the full force-field level was calculated. The contextualization of motor memory based on decision uncertainty predicted a significant difference in coefficients between the two fields. However, single-context learning predicted no differences between the two groups.

For Experiments 2-2 and 3-3, to observe the effect of congruency (match of direction between the motion and target), we also compared the force compensation level between the congruent and incongruent conditions across the two uncertainty contexts (Supplementary Fig. S2m,n).

**Comparing the effect across different conditions in Experiments 2 and 3.** To quantify and compare the effects across the experiments, we simply calculated the difference in the force compensation level between the strong and weak force conditions in Experiment 2 (dual-magnitude force fields), and between the certain and uncertain conditions in Experiment 3 (opposing force fields). This value represents the degree of separation of the motor memory tagged by the context.

**Data analysis of Experiment 4.** The data were analysed in a manner similar to that in Experiment 2. The analysis was conducted separately for the random-dot motion and arrow sequence stimuli. The correspondence between the contextual effects of the random-dot motion and the arrow stimuli was assessed by calculating the difference in the force compensation level between the two force-field conditions for each participant and plotting them against each other (correlation) (Fig. 4e).

**Data analysis of Experiment 5.** Force compensation during the error-clamp probe trials of the first reaching was analysed. Trials with double-step (follow-through) reaching (GO trials) and NOGO trials were analysed separately. For the NOGO trials, participants who moved >2 cm from the central target were excluded from the analysis.

### Statistical analysis
For Experiment 1, a two-way ANOVA (group (across participant factor 2) × coherence level (within-participant factor 5)) and *t*-test (repeated measurement) were used for the statistical test. Unless specfied, a two-sided *t*-test (repeated measurement) was used for all pairwise comparisons and the Bonferroni method was used to correct for multiple comparisons.

### Reporting summary
Further information on research design is available in the Nature Portfolio Reporting Summary linked to this article.

## Data availability
All data required to evaluate the conclusions of the study are presented in the paper and in the Supplementary Information, and have been deposited on the OSF website (https://osf.io/n7z4q/)[48].

## Code availability
The code needed to reproduce all the figures has been deposited on the OSF website[48] together with the data.

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

## Acknowledgements

We thank M. Koshimizu for help in the data collection process; members of the CiNet Motor Control Unit, HONDA R&D, and J. Heald for helpful insights during discussions. Part of this study was supported by grants from the Japan Society for the Promotion of Science (Kakenhi:20H00107, 21H00314) and the Japan Science and Technology Agency (ERATO: JPMJER1801) to N.H. The funders had no role in study design, data collection and analysis, decision to publish or preparation of the manuscript.

## Author contributions

N.H. conceived the study. K.O., A.Y., G.O., M.N., M.H. and N.H. designed the experiments. K.O. and N.H. collected the data. K.O., A.Y., G.O., M.H. and N.H. analysed the data. N.H. wrote the manuscript. K.O., A.Y., G.O., M.N., M.H. and N.H. reviewed and edited the manuscript.

## Competing interests

M.N. is an employee of Honda R&D Co. Ltd. The other authors declare no competing interests.

## Additional information

**Correspondence and requests for materials** should be addressed to Nobuhiro Hagura.

ns licence, unless indicated otherwise in a credit line to the material. If material is not included in the article's Creative Commons licence and your intended use is not permitted by statutory regulation or exceeds the permitted use, you will need to obtain permission directly from the copyright holder. To view a copy of this licence, visit http://creativecommons.org/licenses/by/4.0/.

© The Author(s) 2024, corrected publication 2024

# Reporting Summary

## Statistics

For all statistical analyses, confirm that the following items are present in the figure legend, table legend, main text, or Methods section.

| n/a | Confirmed | |
|---|---|---|
| ☐ | ☒ | The exact sample size (*n*) for each experimental group/condition, given as a discrete number and unit of measurement |
| ☐ | ☒ | A statement on whether measurements were taken from distinct samples or whether the same sample was measured repeatedly |
| ☐ | ☒ | The statistical test(s) used AND whether they are one- or two-sided<br>*Only common tests should be described solely by name; describe more complex techniques in the Methods section.* |
| ☐ | ☒ | A description of all covariates tested |
| ☐ | ☒ | A description of any assumptions or corrections, such as tests of normality and adjustment for multiple comparisons |
| ☐ | ☒ | A full description of the statistical parameters including central tendency (e.g. means) or other basic estimates (e.g. regression coefficient) AND variation (e.g. standard deviation) or associated estimates of uncertainty (e.g. confidence intervals) |
| ☐ | ☒ | For null hypothesis testing, the test statistic (e.g. *F*, *t*, *r*) with confidence intervals, effect sizes, degrees of freedom and *P* value noted<br>*Give P values as exact values whenever suitable.* |
| ☒ | ☐ | For Bayesian analysis, information on the choice of priors and Markov chain Monte Carlo settings |
| ☒ | ☐ | For hierarchical and complex designs, identification of the appropriate level for tests and full reporting of outcomes |
| ☐ | ☒ | Estimates of effect sizes (e.g. Cohen's *d*, Pearson's *r*), indicating how they were calculated |

*Our web collection on statistics for biologists contains articles on many of the points above.*

## Software and code

Policy information about availability of computer code

| Data collection | A haptic device (PHANToM Premium 1.5 HF) was programed by C++ (Visual studio version 2008) to be used as a manipulandum to collect the reaching data. |
|---|---|
| Data analysis | All the data analysis was carried out using Matlab version 2020b (Mathworks). Data frame toolbox (https://www.diedrichsenlab.org/toolboxes/matlab_toolboxes.htm) was used for creating the figures. Code to reproduce the figures to interpret the data in the paper have been deposited on the OSF website (https://osf.io/n7z4q/), along with the data. |

For manuscripts utilizing custom algorithms or software that are central to the research but not yet described in published literature, software must be made available to editors and reviewers. We strongly encourage code deposition in a community repository (e.g. GitHub). See the Nature Portfolio guidelines for submitting code & software for further information.

## Data

Policy information about availability of data

All manuscripts must include a data availability statement. This statement should provide the following information, where applicable:
- Accession codes, unique identifiers, or web links for publicly available datasets
- A description of any restrictions on data availability
- For clinical datasets or third party data, please ensure that the statement adheres to our policy

All the data required to evaluate the conclusions of the study are presented in the paper and in the Supplementary Materials, and have been deposited on the OSF website (https://osf.io/n7z4q/).

## Research involving human participants, their data, or biological material

Policy information about studies with human participants or human data. See also policy information about sex, gender (identity/presentation), and sexual orientation and race, ethnicity and racism.

| | |
|---|---|
| Reporting on sex and gender | 46 female and 101 male volunteers participated in the study (self-reported). Sex and gender were not considered in the study design, since we lack specific hypothesis of any group difference regarding the effect of interest. Therefore, any sex- and gender-based analysis is not performed. |
| Reporting on race, ethnicity, or other socially relevant groupings | Participants were not classified into different race, ethnicity of other social categories. |
| Population characteristics | Participants were recruited from the participant pool of CiNet (mainly university students and researchers), ranging in age from 19-38 years old. All subjects were right-handed. |
| Recruitment | All participants were recruited at CiNet via an online system (SONA systems) for volunteers, for the compensation of 1000JPY/hour for participation. |
| Ethics oversight | National Institute of Information and Communications Technology (NICT) ethical committee |

Note that full information on the approval of the study protocol must also be provided in the manuscript.

# Field-specific reporting

Please select the one below that is the best fit for your research. If you are not sure, read the appropriate sections before making your selection.

☒ Life sciences ☐ Behavioural & social sciences ☐ Ecological, evolutionary & environmental sciences

For a reference copy of the document with all sections, see nature.com/documents/nr-reporting-summary-flat.pdf

# Life sciences study design

All studies must disclose on these points even when the disclosure is negative.

| | |
|---|---|
| Sample size | Participants were randomly sampled and assigned to each experiment. We did not pre-define the sample size. While sample sizes for experiment groups in analogous motor learning studies typically range between 8 and 12, here, we used larger sample sizes in Experiment 1 (N = 19 each for both the certain and uncertain condition) due to the cross-participant design, and also to account for the possible noise induced by the trial-by-trial fluctuation of the subjective uncertainty level within participants. To ensure a similar level of effect size as in Experiment 1, we used a similar number of participants in the rest of the experiments (Experiment 2-1, n=19; 2-2, n=20; 2-3, n=17; 3-1, n=18; 3-2, n=16; 3-3, n=15; 4, n=18; 5, n=14) |
| Data exclusions | In each experiment, trials were excluded if the 1) reaction times (movement onset concerning the visual stimulus onset) were too fast (<100 ms; likely not judging the stimulus) or too slow (1,500 ms>; judging after the stimulus disappearance), 2) did not reach properly to the target (<75% of the maximum distance), and when the movement direction reversed after going 2.5 cm to the opposite direction before reaching to the target. If the trial exclusion rate exceeded 30% of the data in the last block of the learning phase or the retrieval/test phase, the participants were excluded from further analysis. Furthermore, if the overall choice rate during the retrieval phase was biased towards one direction (>70%) (e.g., moving [making a decision] to the right in most of the trials), the participant was also excluded because of the asymmetrical motor learning experience between the two directions. See the method section below for task details. Note that these exclusion criteria were set to exclude data/participants who did not follow the instructions of the experiments and maintain the same data quality across participants. However, including excluded participants in the analysis did not qualitatively change the results.
Based on the above criteria, in Experiment 1, three participants from each certain and uncertain group were excluded. Likewise, two participants were excluded from the analysis of Experiment 2-3, 2-4, and 3, respectively. |
| Replication | Result of Experiment 1 was conceptually replicated in Experiment 2-1 and 2-2. Experiment 3 also included the replication of Experiment 1. |
| Randomization | Participants were randomly assigned to each experiment. |

| Blinding | Experimenter was not blind to the purpose of the experiment. However, the experimenter could not always monitor the condition of the current trial, due to the location they sat during the experiment. Furthermore, the effect was also replicated (Experiments 3-1, 3-2) when different experimenter who were blind to the purpose of the study conducted the experiment. |
|---|---|

# Reporting for specific materials, systems and methods

We require information from authors about some types of materials, experimental systems and methods used in many studies. Here, indicate whether each material, system or method listed is relevant to your study. If you are not sure if a list item applies to your research, read the appropriate section before selecting a response.

## Materials & experimental systems

| n/a | Involved in the study |
|---|---|
| ☒ | Antibodies |
| ☒ | Eukaryotic cell lines |
| ☒ | Palaeontology and archaeology |
| ☒ | Animals and other organisms |
| ☒ | Clinical data |
| ☒ | Dual use research of concern |
| ☒ | Plants |

## Methods

| n/a | Involved in the study |
|---|---|
| ☒ | ChIP-seq |
| ☒ | Flow cytometry |
| ☒ | MRI-based neuroimaging |

