## [Peer Review File · Nature Human Behaviour]

Peer Review Information

Journal: Nature Human Behaviour

Manuscript Title: Decision uncertainty as a context for motor memory

Corresponding author name(s): Nobuhiro Hagura

Reviewer Comments & Decisions:

Decision Letter, initial version:

17th May 2023

Dear Dr Hagura,

Thank you once again for your manuscript, entitled "Decision uncertainty as a context for motor memory", and for your patience during the peer review process.

Your article has now been evaluated by 3 referees. You will see from their comments copied below that, although they find your work of potential interest, they have raised quite substantial concerns. In light of these comments, we cannot accept the manuscript for publication, but would be interested in considering a revised version if you are willing and able to fully address reviewer and editorial concerns.

We hope you will find the referees' comments useful as you decide how to proceed. If you wish to submit a substantially revised manuscript, please bear in mind that we will be reluctant to approach the referees again in the absence of major revisions. We are committed to providing a fair and constructive peer-review process. Do not hesitate to contact us if there are specific requests from the reviewers that you believe are technically impossible or unlikely to yield a meaningful outcome.

To guide the scope of the revisions, the editors discuss the referee reports in detail within the team, including with the chief editor, with a view to (1) identifying key priorities that should be addressed in revision and (2) overruling referee requests that are deemed beyond the scope of the current study. We hope that you will find the prioritized set of referee points to be useful when revising your study. Please do not hesitate to get in touch if you would like to discuss these issues further.

Specifically, we ask you to [1] perform the control experiment requested by Reviewer #2, [2] address the technical issues raised by Reviewers #1 and #3, and [3] discuss how the recent and impactful COIN-model fits the present results (Reviewer #1). Additionally, we are concerned by the comments of Reviewers #1 and #3 about the limited and variable extent of adaptation (comments 2 and 1,

respectively). If you wish to resubmit a revised version of this manuscript, [4] you must fully and convincingly address these concerns with additional data and analyses. Finally, [5] please ensure that the link to your data and code is active and can be accessed anonymously, so that the reviewers could verify their analyses (comment 2 by Reviewer #1).

Your revised manuscript must comply fully with our editorial policies and formatting requirements. Failure to do so will result in your manuscript being returned to you, which will delay its consideration. To assist you in this process, I have attached a checklist that lists all of our requirements. If you have any questions about any of our policies or formatting, please don't hesitate to contact me.

If you wish to submit a suitably revised manuscript, we would hope to receive it within 4 months. I would be grateful if you could contact us as soon as possible if you foresee difficulties with meeting this target resubmission date.

- Include a "Response to the editors and reviewers" document detailing, point-by-point, how you addressed each editor and referee comment. If no action was taken to address a point, you must provide a compelling argument. When formatting this document, please respond to each reviewer comment individually, including the full text of the reviewer comment verbatim followed by your response to the individual point. This response will be used by the editors to evaluate your revision and sent back to the reviewers along with the revised manuscript.
- Highlight all changes made to your manuscript or provide us with a version that tracks changes.

[REDACTED]

Thank you for the opportunity to review your work. Please do not hesitate to contact me if you have any questions or would like to discuss the required revisions further.

Sincerely,

[REDACTED]

Reviewer expertise:

Reviewer #1: Motor control, computational neuroscience

Reviewer #2: Motor learning, memory consolidation

Reviewer #3: Motor control, sensorimotor adaptation

REVIEWER COMMENTS:

Reviewer #1:

Remarks to the Author:

Decision uncertainty as a context for motor memory

Ogasa, Yokoi, Okazawa, Nishigaki, Hirashima and Hagar

Summary:

This study presents a well designed set of experiments, showing that decision uncertainty can serve as a context for motor memory. First, the authors present in experiment where subjects have to make reaches through a force field, either to a left or right target. The target direction is indicated by an RDM stimulus. In one group this stimulus has high (100%) coherence, in the other group it has low (3%) coherence. After training, the authors probe generalisation across different coherence levels and show that compensation declines as the coherence moves away from the training coherence. So the high uncertainty (3%coh) shows higher compensation for this low coherence than for the highest. Next, the authors investigate learning to compensate for two opposing force fields that are intermixed. Again, subjects view an RDM stimulus that indicates whether to reach for the left or right target, but the coherence level indicates the magnitude or direction of the force field. This experiment is done both with different magnitudes and with different force field directions. As a control, to see whether it is the coherence perse or in combination with the decision making process, the authors present, for magnitude only, that coherence alone does not result in dual learning. Finally, the authors present an experiment in which they show that it is uncertainty, not the perceptual stimulus itself that results in dual learning. This was again for learning force field of different magnitude, not sign.

Evaluation:

This paper addresses the relevant and timely question whether uncertainty plays a role in the formation of (multiple) motor memories. The authors address this by looking at generalisation of a learned motor memory along the uncertainty dimension and by looking at learning of two different force field simultaneously, where uncertainty serves as a context.

Although the introduction and paper are well written, I believe that the abstract should be completely revised. I could not guess what you were after after reading this and even after having read the paper I had a hard time linking it to the abstract. For a general audience it should be more accessible. One sentence that stood out from the intro, and should be basis for abstract is: "... decision uncertainty works as a contextual cue for motor memory." Also the sentence (ln 146): "... the incomplete transfer of motor memory across different decision uncertainties implies that the decision process preceding the action can be a context for motor memory", would be a good sentence for the abstract.

Furthermore there are some technical details, hidden in the paper, that make the result not as clean as I would have hoped it to be. First, the force fields are relatively weak, especially in study 2. The adaptation to these force field also seems rather limited. I would like to see more of the individual data instead of bar graphs and SDs. I tried to open the OSF link, but that is not yet active. Wanted to see for myself the variance that is behind these data.

Third, the authors use a random dot motion paradigm that may not be as 'uncertain' as implied. The subjects can track these dots very well.

So, I am sympathetic to the question addressed and the experimental paradigms are well thought of. However, some choices in the paper make me wonder how much these findings depend on the specific choices made by the authors. Many of these choices are different than what I would have expected (type of RDM, way of computing the learning index (single point), normalising the outcomes, magnitudes of force fields).

As a final remark, I am curious how the authors interpret their findings in terms of the COIN-model as recently published by Heald, Lengyel and Wolpert. It would be good to discuss how contextual inference (here uncertainty) models could explain the present findings. I know this would be speculation, but it would give the paper a broader scope. I always loosely summarise that contextual cues need to be within the sensorimotor loop to be effective. As a result, color does not work, but follow-thru movements do. Curious how the current findings would change this interpretation.

Detailed:

Pg 2: In 60: 'similar' and 'similarly' in the same sentence. Please avoid.

Pg 2: In 63-65: I wrote 'massive coherence difference'. Later I discovered in the methods that the dots do not follow the standard 'Newsome' implementation and as a result may not be that different. The present dots you can track, as they have no limited lifetime. These dots do not really ask for "perceptual decision making". I don't want you to redo the experiments, but I would like a reflection. Especially because also in the arrow task at the end you have a 'counting task', not necessarily an uncertainty task.

Fig 1. Learning coefficient in panel D is probably 'adaptation index', right? Just to be sure. However, why do you switch in panel E to retrieval ratio? I would like to see this graph in learning coefficient or adaptation index. This may also show why the 'certain' group still improves after training. How did you quantify 'amount of force learned at the end of the learning phase'? How many channels were included?

Ln 115-116: consider: "..., we predict the best retrieval performance at the same level of decision uncertainty in which the motor memory is formed."

Ln 127: Not sure what statistical analysis you performed here. This is a mixed level analysis, right? You compare slopes across groups. Or what does your ANOVA (and especially the interaction) look like?

Ln 138: effect(s) of perceptual learning -> clarify

Ln 138-141: these sentences do not flow and are vague. Please rewrite

Ln 142: I know ref 19, but for most readers this will be abracadabra. Please give away a bit more about the control.

Fig 2 panel C and D: Learning coefficient curves. I would like to see data from the learning phase as well. Learning coefficients are rather low, given that for standard FFs we see >80% compensation. Why

are your learning coefficients so low? And, I find the variance at the asymptote in panel D disturbing. Did the subject really learn something here?

You perform nice controls in Fig S3, but you subjects still learned to compensate for the average force field. Unfortunately you only control for the magnitude experiment, but with the opposing force field no learning should happen at all. I would like to see this control! Also, were the direction random from a set of two (so congruent vs incongruent) or were they truly in random directions? I ask because if the first is the case, you have a confound. (see Ln 481, which is ambivalent)

Ln 204: 'This result' -> seems a remnant from a previous version. There is no result before.

Ln 216: mean is 0.23, but how do I relate that to the 0.5 vs 0.4 for exp 2.1 in the previous paragraph?

Fig. 2E: Expected difference ratio, I find this not a very intuitive measure.

Ln 382-388: compare to the 'banks' that Newsome and Shadlen use.

Ln 389 and onwards: what do you define as movement initiation.

curl fields: exp 1: $b = 10$ Ns/m, exp 2.1: $b = 10$ Ns/m vs 5Ns/m, exp 2.2: 2.5Ns/m -> especially the latter one you will need to defend! You reduced by a factor 4 from the original! Why? This is not a random choice. Only the 10 Ns/m approaches the values you generally see in the literature (e.g. Forano, 2020)

Ln 522: This sentence does not flow and I don't understand what you are after.

Ln 571: Learning coefficient based on a single point from the curve? That is far from standard. Does it matter whether you perform a regression without offset? I.e. more data into a single number.

Ln 578: Ah, here is the retrieval ration. Last block, is how many trials?

Ln 580-593: You describe the model, but please also provide the fitting procedure and the values that came out of it. How did you construct the average curve?

Ln 606: put 'expected difference ratio' in heading. Makes reading much easier. However, I also prefer a different naming here. It is not the predicted difference ratio, but the ration between the predicted and actual difference between conditions. A value of one indicates perfect performance. This paragraph is critical to understand many of the figures.

This also makes me wonder, again, what I am looking at in figure 2E: if I understand your measure correctly, perfect performance would be 1.0, but you hardly touch 0.25. Can you really claim that they 'learn'? Same holds for figure 3.

Reviewer #2:

Remarks to the Author:

Ogasa and colleagues conducted a series of experiments to investigate the role of context uncertainty

in tagging motor memories. To this aim they performed a series of experiments in which they manipulated the level of certainty/uncertainty of the context in which a decision to move in a force field was made based on the level of congruency of perceptual cues. Experiment 1 showed that retrieval of the force was highest when the level of uncertainty matched that of the cue during encoding. Experiment 2 demonstrated that the level of uncertainty provided by the cue enabled simultaneous learning of two different force fields. In Experiment 3, the authors examined the relevance of stimulus features during encoding for memory tagging. Based on these experiments the authors concluded that the level of certainty during decision making can act as a cue to tag different motor memories.

I find the study to be interesting and relevant to the field. The experimental manipulation is clever, simple and original. Overall, the experimental design and data analysis are sound. My major concern relates to the interpretation of the results. The authors suggest that their findings reflect a modulation of the decision process. However, I am not convinced that they provide sufficient evidence to support this hypothesis. The authors suggest that the visual cues used in the study are weak contextual cues, and thus unlikely to act as a tag on their own to encode different memories. However, the stimuli used to manipulate the level of uncertainty in which movements are planned are dynamic or imply motion, and such stimuli have been shown to tag different motor memories when the cued movement takes place within a time window of less than 600 ms (Howard et al., 2012; Howard et al., 2013). I understand that Experiment 2.3 was designed to distinguish between the role of contextual uncertainty and the influence of stimulus features in producing the effects observed in Figures 1 and 2. However, a potential issue with this experimental design is that the cue and the resulting movement may not correspond, which could lead to reduced levels of sensorimotor tagging. Experiment 3 also uses a sequence of arrows which presumably implies motion, further suggesting the possibility of a sensorimotor confound. To test the hypothesis that actions can be tagged by the certainty/uncertainty of the decision context itself, a static and preferably abstract cue, irrelevant to movement execution, should be used to code the level of uncertainty. Alternatively, an interval longer than 600 ms between the presentation of the cue and movement initiation could be imposed to discard a sensorimotor confound.

The figures in the manuscript require additional work to ensure that readers can understand them by reading the captions alone. In many instances, the information provided in the figure does not match the descriptions in the Results or Methods section. For example, Figure 1.B suggests that only the retrieval phase includes error-clamp trials, while the learning phase does not. Similarly, Figure 1.C specifies probe trials only during the retrieval phase, which contradicts the information provided in the Methods. To clarify the experimental design, it would be helpful to merge Figure 1.A and Supplementary Figure 1 so that readers can see both the display and the experimental setting. Additionally, in Figure 3, the cue changes, but the display is not the same as in Figure 1, even though the text suggests it should be. To address this, it may be useful to include a replica of Figure 1.A, but with an arrow replacing the dots. Please clarify in the caption of Figure 1 what the error bars indicate. Overall, the figures would benefit from additional effort to ensure consistency and clarity.

The description of Experiment 1 lacks clarity, which makes it difficult for the reader to understand the methodology and the results. The information provided in the Results section is incomplete and inconsistent with what is presented in Figure 1. Although the caption mentions two phases of the experiment, "learning" and "retrieval," the text fails to elaborate on them adequately. The figure suggests that the retrieval phase solely consists of probe trials, but the Methods section states that the retrieval phase is the same as the learning phase except for the type of probe trials. Additionally,

the Methods section lacks important details, such as whether there were any breaks within each phase and if so, their duration and location. The Methods section also does not specify whether there were any breaks between the learning and retrieval phases or how long it took to complete the entire task.

I have a concern regarding the distinction between the learning and retrieval phases in this study. According to the Methods, the only difference between these phases is the type of error-clamp trials used. However, it is not clear why the second phase is labeled as "retrieval." Figure 1D suggests that subjects have already reached asymptotic performance by the end of the learning phase. If there is no break between the two phases, then the so-called "retrieval" phase could be interpreted as overlearning, i.e., continued training at the asymptote. Additionally, I am unclear why only the probe trials from the retrieval phase are considered to measure memory retention/retrieval. How do the two phases differ beyond asymptotic performance? If the goal was to measure memory retention/retrieval after learning, why not conduct a separate test session after asymptotic performance was reached, without further learning?

The Discussion could be extended to elaborate on how this study differs from previous literature addressing the impact of context on motor memory tagging. The work by Song and collaborators (2013, 2015) showing that visuomotor adaptation memories acquired in a context of attentional distraction are better recalled under a similar level of distraction than under a no-distraction context, is very relevant to this study but is not mentioned neither in the Introduction nor in the Discussion. The authors could also consider whether similar mechanisms may be at play in the motor system's use of different strategies for motor control in the face of uncertainty, as suggested in previous literature (e.g., Chouinard and Paus, 2005).

Finally, I have some minor comments. Please clarify whether "nonerror-clamp trials" refer to regular learning trials in which no clamp is applied (line 432). Lines 129-141 could be reworded for clarification.

Reviewer #3:

Remarks to the Author:

Hagura and colleagues test whether the uncertainty of a decision to move right or left acts as a context that affects the formation of multiple motor memories. In a series of 3 related experiments (and additional control experiments), they manipulated the decision uncertainty by changing the coherence of a moving dot (or arrows) such that some groups were exposed to 100% certain direction while the others were exposed to less certain direction (e.g., 3.2%), during adaptation to external velocity-dependent force field perturbations. Adaptation has been tested in multiple conditions of force-fields with different magnitude (strong vs weak) but similar direction (CC or CCW), or force field with opposite directions but similar magnitude. Their approach aims at distinguishing between the retrieved motor memories by using channel trials during the late phase of learning, under each uncertainty condition.

The main finding of the study is that when participants were required to judge the direction of the moving dots, and then commit to this direction, this decision process allowed them to differentiate between different environments. Practicing the same visual uncertainty without making a decision, or changing the time to reach the decision, has much smaller effect on the ability to retrieve the correct response.

Overall, this is a decent amount of work, and the 3 experiments (and the controls) are logically linked and address a related hypothesis. However, there are several shortcomings that should be addressed in detail. In addition, the speculation that planning-related mechanism underlies the covert internal decision is a testable question and my suggestion is to perform this experiment to strengthen the main finding (see my comments below).

1. There is inconsistency in the amount of learning across the different experiments. To this reviewer it is not clear why participants reached different learning levels across the different experiments. For example, in exp. 1, participants reached a level of ~65-70% of the ideal force in the different uncertainty conditions, while in the same condition (e.g., strong FF and certain) in exp. 2-1, participants reached at maximum 55%, and ~50% in exp. 3 for the same condition. At first, I thought that the learning phase was shorter in these experiments but when we look at the late phase of the learning curve (Fig., 2C), it looks like the participants reached asymptote, suggesting that something else beyond the length of the learning phase, affected the learning process. Do the authors have any explanation for what might cause this? Also, I am curious as to whether some interference could have happened during the switch between the FFs.
2. The quantification of the learning using the learning coefficient based on a single point of the force profile (at max velocity) is unclear to this reviewer and seems to be very sensitive to outliers in the data. Typically, to give a robust measure of learning, the actual forces in the channel's trials should be regressed with the ideal force, and then the regression coefficient is used to determine how much the participant learns from the ideal FF. This robust regression is immune to any unstable movements that might occur. I suggest conducting the analysis of learning, but instead of focusing on a single data point, perform the regression across the entire force profile. Related, the paper is largely lacking information about the force profiles across learning and retrieval. This should be at least added to the Supplementary Information.
3. The speed of movement across the different conditions is lacking in all experiments. This analysis should be carefully conducted, and the results should appear in the manuscript. The speed of a movement can hint to the learning mechanism during the learning phase (i.e., feedforward vs. feedback). The fact that the authors analyzed the force profile at a single point and only during the force channel trials do not provide deep understanding of the nature of the learning.
4. The speculation that uncertainty of the decision forms a context that might affect premovement (e.g., planning) is very interesting, but direct evidence in this paper is lacking. Nevertheless, since this is a testable prediction using a rather simple behavioral paradigm (see for example the follow-through experiments of opposing FFs in previous work), I highly recommend performing this experiment and contrasting planning vs. execution while manipulating the uncertainty of the decision. This would confirm the main conclusion, and most importantly, confirm the speculated mechanism.
5. Generalization of learning from stronger to weaker FF (and perhaps vice versa) is highly possible. If this is the case, then the amount of learning reported when changing the context from high to low uncertainty (or the opposite) might be also related to the generalization between the two environments. This should be somehow quantified and further discussed.
6. The timing of the decision process might also affect the learning. It is not clear (perhaps the reviewer missed it) if there is a correlation between the decision time (time from dot's motion onset to movement onset) and the learning amount and whether there were differences between the different uncertainty conditions/groups.

Secondary comment:

Fig. 1E. it is very puzzling as to why participants in the Uncertain condition would learn less if they are exposed to a 100% certainty of where to move. How was this expressed? Is it in a form of lower force profile compared to learning uncertain condition? Also, I am curious about comparing the retrieval ratio across groups but here matching the uncertainty during learning. That is the retrieval for the uncertain group during 100% and the certain group in the 3% uncertainty. This would tell us whether the retrieval is purely context dependent or there is something else occurring during the decision process.

Author Rebuttal to Initial comments

POINT-BY-POINT REPLY LETTER

Revised sections are highlighted in red in the manuscript.

Reviewer #1

Comment 1: This paper addresses the relevant and timely question whether uncertainty plays a role in the formation of (multiple) motor memories. The authors address this by looking at generalisation of a learned motor memory along the uncertainty dimension and by looking at learning of two different force fields simultaneously, where uncertainty serves as a context.

Reply: We appreciate the overall positive evaluation to our paper and the constructive suggestions to improve our paper!

Comment 2: Although the introduction and paper are well written, I believe that the abstract should be completely revised. I could not guess what you were after after reading this and even after having read the paper I had a hard time linking it to the abstract. For a general audience it should be more accessible. One sentence that stood out from the intro, and should be basis for abstract is: "... decision uncertainty works as a contextual cue for motor memory." Also the sentence (ln 146): "... the incomplete transfer of motor memory across different decision uncertainties implies that the decision process preceding the action can be a context for motor memory", would be a good sentence for the abstract.

Reply: Thank you very much for the suggestions. We have now revised the abstract as below, also reformatting the introduction. We hope the novelty of the paper is clearer with this.

<Abstract>

The current view of perceptual decision-making suggests that once the decision is made, a single motor program associated with the decision is carried out, irrespective of the uncertainty

involved in the decision-making. As opposed to this view, we show that multiple motor programs can be acquired based on the preceding uncertainty of the decision, indicating that decision uncertainty works as a contextual cue for motor memory. Actions learned following certain (uncertain) decisions only partially transferred to uncertain (certain) decisions. Participants were able to form distinct motor memories for the apparently same movement based on the preceding decision uncertainty. Crucially, such contextual effect generalized to novel stimuli with matched uncertainty levels, demonstrating that the decision uncertainty itself is the contextual cue. Our findings broaden our understanding of contextual inference for motor memory, emphasizing that it extends beyond the direct motor control cues to encompass the state of the decision-making process.

Comment 3: Furthermore there are some technical details, hidden in the paper, that make the result not as clean as I would have hoped it to be. First, the force fields are relatively weak, especially in study 2. The adaptation to these force field also seems rather limited.

Reply: Thank you very much for pointing this out.

Regarding the force-field strength, we have re-performed the conflicting-force experiment by doubling the strength of the force-field to the original. We were able to obtain qualitatively (statistically) equivalent level of results. This experiment has been now added to the revised manuscript (new Experiment 3-2).

Regarding the level of force-field compensation, we acknowledge that the amount of adaptation does not fully compensate for the force-field. Nevertheless, we still do think this adaptation level is significant to claim the motor memory separation by the decision uncertainty context for the following reasons.

As we have now discussed in the revised manuscript, the key distinction of the current findings from previously reported contextual cues for motor memory is that the cue (i.e. decision uncertainty) is not a direct sensory input that specifies the action context, but the cue itself is a product of an inference about the stimulus (refer also to our reply regarding the COIN model). This is confirmed by the result of Experiment 4, where we demonstrated that the motor memory can be tagged to a decision uncertainty level of the stimulus, independent of the physical feature of the visual input. In such situation, while we can quantify the overall effect of uncertainty by analysing choice patterns, the trial-by-trial subjective uncertainty level can vary. For instance, when deciding the motion direction for a 3% coherent motion stimulus, confidence may vary across trials, sometimes confident, sometimes not. Despite this, we consistently observed a statistically strong effect (effect size; $d_z > 1.0$) of decision uncertainty context across all the experiments throughout the study, including the additional experiments in the revised manuscript (new Experiments 3-2, 5). Furthermore, the average compensation level was also comparable to that reported in a previous study (Howard et al., 2015), especially when a challenging combination of contexts (follow-in, follow-through contexts) were associated with different force-field patterns (i.e.,

compensation level at the end of Day 1; <20%). Therefore, our findings indicate that the contextual inference process can utilise the inferential state of the environment, thus can go beyond the sensory cues that directly specifies the action. Such characteristics of input fluctuations inherent to the uncertainty inference may necessarily limit the contextual effect based on decision uncertainty.

In summary, we assert that decision-uncertainty is indeed a significant context for motor memory. This point is now described in the discussion section, page 13, lines 9-33.

Comment 4: I would like to see more of the individual data instead of bar graphs and SDs. I tried to open the OSF link, but that is not yet active. Wanted to see for myself the variance that is behind these data.

Reply: Now, we have provided individual points on each figure. Force profile is now appears in Supplementary Materials, Fig. S4. Also, OSF link (<https://osf.io/n7z4q/>) is now activated, and the related data is publicly available. Please refer to the Figures and the OSF link.

Comment 5F: So, I am sympathetic to the question addressed and the experimental paradigms are well thought of. However, some choices in the paper make me wonder how much these findings depend on the specific choices made by the authors. Many of these choices are different than what I would have expected (type of RDM, way of computing the learning index (single point), normalising the outcomes, magnitudes of force fields).

Reply: Thank you very much for pointing them out. We have now revised the description and the analysis, according to the reviewer's comment.

- We apologise for the incomplete description of the RDM parameters. RDM we used had dots with a limited lifetime (133ms). Thus, participants cannot keep tracking the trajectory of one single dot to decide the movement direction. We have now revised the description. Please see the Method section, page 16, lines 4-12.
- All the learning indices have been now re-calculated and replaced by the standard regression method. Our claim was unaffected by changing the analysis. Please see the Result and the Method section. page 22, lines 3-8.
- Following your advice, we have re-performed the conflicting-force experiment with stronger force-field (Experiment 3-2) (also with the matched control experiment; Experiment 3-3), and again, we were able to obtain comparable amount of effect. Please see the result of new Experiments 3-2 & 3-3 and Fig. 3.

Taken together, the decision uncertainty can indeed be a contextual cue for motor memory, not confined to a very particular situation.

Comment 6: As a final remark, I am curious how the authors interpret their findings in terms of the COIN-model as recently published by Heald, Lengyel and Wolpert. It would be good to discuss how contextual inference (here uncertainty) models could explain the present findings. I know this would be speculation, but it would give the paper a broader scope. I always loosely summarise that contextual cues need to be within the sensorimotor loop to be effective. As a result, color does not work, but follow-thru movements do. Curious how the current findings would change this interpretation.

Reply: Thank you very much for the intriguing question. COIN model (Heald et al., 2021) assumes that the learner infers contexts based on observations of the external world (e.g. sensory cues). However, in our experiment, we show that the learner uses decision uncertainty, which is independent from the sensory cue, to infer contexts. Thus, we found that uncertainty reduces interference between memories, whereas in the COIN model, uncertainty (about the context at least) increases interference. Such internal, inference-based contextual cue is “*not assumed in the current version of the COIN model*” (personal communication with James Heald). Therefore, our finding broadens our understanding of contextual dependent motor memory, extending the scope of the contextual cues to inferential states, such as the decision uncertainty.

Our study does not deny the involvement of the sensorimotor process in the decision-uncertainty context, since decision must be expressed through action, and the sensorimotor state continuously tracks the ongoing decision (Cisek & Kalaska, 2010, Gold and Shadlen, 2000). The novel point we have demonstrated is that the internally inferred *quality* (uncertainty) of the decision, which seems separated from the executed action, once the decision has been made, can still be a context for motor memory. Please also refer to our new Experiment 5 (Fig. 5), which strengthens our claim.

These points are now highlighted in the Discussion section, Page 13, lines 9-23.

Comment 7: Pg 2: ln 60: 'similar' and 'similarly' in the same sentence. Please avoid.

Reply: Thank you for the suggestion. Has been now revised.

Comment 8: Pg 2: ln 63-65: I wrote 'massive coherence difference'. Later I discovered in the methods that the dots do not follow the standard 'Newsome' implementation and as a result may not be that different. The present dots you can track, as they have no limited lifetime. These dots do not really ask for "perceptual decision making". I don't want you to redo the experiments, but I would like a reflection. Especially because also in the arrow task at the end you have a 'counting task', not necessarily an uncertainty task.

Reply: We apologise for the missing detail of the stimulus parameters. The dots indeed have lifetime of 133msec, thus is a standard RDM stimulus. We have now revised the description of the stimulus. Please see page 16, lines 4-12 in the Methods section.

Comment 9: Fig 1. Learning coefficient in panel D is probably 'adaptation index', right? Just to be sure. However, why do you switch in panel E to retrieval ratio? I would like to see this graph in learning coefficient or adaptation index. This may also show why the 'certain' group still improves after training. How did you quantify 'amount of force learned at the end of the learning phase'? How many channels were included?

Reply: Sorry for causing confusion by using different indices. The use of retrieval ratio was to intuitively show the amount of recall related to what the participants have learned. We have now renamed it to generalization in Figure 1E, since we are evaluating the generalisation of the motor memory learned in one uncertainty level to other decision uncertainties. Also, we have now provided the force compensation data, before converting it to generalisation score, in the Supplementary Fig. S1. Force compensation is now calculated using the regression method (page 22, lines 3-8).

To calculate the generalization of learning to test phase, we used the average compensation during the learning phase calculated from the channel trials of the final block, which includes 12 trials. Is now described in page 22, lines 9-13.

Comment 10: Ln 115-116: consider: "..., we predict the best retrieval performance at the same level of decision uncertainty in which the motor memory is formed."

Reply: Thank you for the suggestion. Is now revised.

Comment 11: Ln 127: Not sure what statistical analysis you performed here. This is a mixed level analysis, right? You compare slopes across groups. Or what does your ANOVA (and especially the interaction) look like?

Reply: Here, we have used two-way ANOVA (Group [2] × Coherence level [6]); one as within participant factor (Coherence levels) and other as across participant factor (Group). These are described in Material and Methods (statistical analysis) section (Page 23, lines 30-34). Statistical significance is now indicated in the figure, which reveals the interaction effect. Please see Fig. 1F

Comment 12: Ln 138: effect(s) of perceptual learning -> clarify

Reply: Now, we have added more detailed explanation about this point as below. Page 5, lines 19-25.

“For the decision-making side, uncertain-decision group slightly increased the visual motion sensitivity and became more conservative than the certain-decision group (Supplementary Fig. S1). This is probably due to the perceptual learning induced by the repeated exposure to the weak

motion signal (20) (21). This tendency was also observed in the pattern of peak-velocity of the movement across different motion coherence levels (Supplementary Fig. S1). However, it is not clear how these results can explain the reversed force production pattern across different decision uncertainty levels between the two groups.”

Comment 13: Ln 138-141: these sentences do not flow and are vague. Please rewrite

Reply: Thank you for the suggestion. We have reconsidered this sentence, and now decided to omit it.

Comment 14D: Ln 142: I know ref 19, but for most readers this will be abracadabra. Please give away a bit more about the control.

Reply: Thank you for the suggestion. We have now rewritten it as follows. Page 5, Lines 9-18.

“It is also unlikely that the feature of the visual stimulus (100% and 3% coherent motion) is the main determinant of this effect. Previous study showed that background colour of the workspace cannot be a contextual cue for motor memory (18). Although, peripheral target rotation direction stimulus has been shown to be a weak contextual cue for motor memory (18) (19), the stimulus used in the present study is motion coherence level independent from the direction. In the below experiments (see below for the results of Experiment 2-2 and 2-6), we further showed that motion coherence level itself (not the decision uncertainty) cannot be a contextual cue for motor memory.”

Comment 15: Fig 2 panel C and D: Learning coefficient curves. I would like to see data from the learning phase as well. Learning coefficients are rather low, given that for standard FFs we see >80% compensation. Why are your learning coefficients so low? And, I find the variance at the asymptote in panel D disturbing. Did the subject really learn something here?

Reply: In the original experiment, unfortunately, we have not included enough error-clamp trials during learning, to maximize learning opportunity. For the new experiments (Experiments 3-2, 3-3, 5), however, we have added error-clamp trials during the learning, and have demonstrated the gradual increase of the force. Please see Fig.3, Fig.5 and Supplementary Fig.S2.

Regarding the amount of compensation level, please see the reply to your comment 3 for detail. We agree that for the standard force-field tasks, >80% compensation level is not a rare thing to observe. However, current task involves competition between two different force fields for the same trajectory, which should interfere with each other. Thus, the situation is quite different to simple force-field reaching tasks. Also, as written above, estimation of uncertainty can fluctuate trial-by-trial, due to the internal fluctuations of the uncertainty estimate, which can disturb establishing the uncertainty context. Finally, the

amount of compensation we observe is comparable to the result of Howard et al., 2015, when a difficult combination of contexts (follow-in, follow-through) were associated with different force-field patterns (i.e. compensation at the end of Day 1; <20%).

The crucial point in the dual force-field learning paradigm is that whether the participants can differentiate the two force-fields according to the given context. Using both magnitude and opposing force paradigms, we have consistently replicated the effect of decision uncertainty context on force-field learning multiple times throughout the study. The evidence in the study, 1) consistent replication of the result with strong effect sizes ($d_z > 1$), 2) comparable amount of learning to the previous studies, points to the conclusion that participants indeed learned to utilise the decision-uncertainty as a contextual cue, to form different motor memories. Now these points are clarified in the discussion section (Page 13. lines 24-33).

Comment 16: You perform nice controls in Fig S3, but you subjects still learned to compensate for the average force field. Unfortunately you only control for the magnitude experiment, but with the opposing force field no learning should happen at all. I would like to see this control! Also, were the direction random from a set of two (so congruent vs incongruent) or were they truly in random directions? I ask because if the first is the case, you have a confound. (see Ln 481, which is ambivalent)

Reply: Thank you very much for the suggestion.

First, the aim of the magnitude experiment is to show that the participants can *differentiate* the two level of forces depending on the given uncertainty context. Thus, generally compensating for the force does not mean that context-dependent compensation is happening. This part is now clarified. Page 6, lines 1-6.

“Suppose the brain can use decision uncertainty to segregate the context and retrieve the relevant motor memory. For the different magnitude force-fields (Experiment 2-1), participants should be able to produce different amount of force depending on the different uncertainty-context. If the uncertainty fails to become a context, although the force-field should be generally compensated in the same direction, difference in the force magnitude level should not be observed across different decision uncertainties.”

Second, as the reviewer requested, we have performed a control condition for the opposing force field (Experiment 3-3). The effect was significantly weaker compared to the corresponding uncertainty condition (Experiment 3-2 vs. 3-3; Fig. 3E and Supplementary Fig. S2). Furthermore, in this control experiment, we also analysed the data by classifying the trials into congruent (direction of the motion and the target matches) and incongruent trials, and again, did not find any difference between these conditions. This shows that simple visual feature of the motion cannot be a strong contextual cue for motor memory, replicating

the previously reported results (Howard et al., 2012). Please see Experiment 3-3 and Supplementary Fig. S2

Comment 17D: Ln 204: 'This result' -> seems a remnant from a previous version. There is no result before.

Reply: We have revised according to the reviewer's suggestion.

Comment 18: Ln 216: mean is 0.23, but how do I relate that to the 0.5 vs 0.4 for exp 2.1 in the previous paragraph?

Comment 19: Fig. 2E: Expected difference ratio, I find this not a very intuitive measure.

Reply: Sorry for the unclear description. We have now decided not to use the 'Expected difference ratio', but instead, just simply take a difference between the % compensation between the two different force fields as an index of motor memory separation induced by the context. Please see Fig. 2 & Fig. 3.

Comment 20: Ln 382-388: compare to the 'banks' that Newsome and Shadlen use.

Reply: We have now added the description about the lifetime of the dots, which were missing in the previous manuscript. We appreciate the reviewer for pointing this out.

Comment 21: Ln 389 and onwards: what do you define as movement initiation.

Reply: Thank you for pointing this out. Movement onset is defined as the point where the tangential velocity of the movement reaches the 10% of the peak velocity. This description has been now added to the Method section. Please see Page 16, line 17-18.

Comment 22: curl fields: exp 1: $b = 10 \text{ Ns/m}$, exp 2.1: $b = 10 \text{ Ns/m}$ vs 5 Ns/m , exp 2.2: 2.5 Ns/m -> especially the latter one you will need to defend! You reduced by a factor 4 from the original! Why? This is not a random choice. Only the 10 Ns/m approaches the values you generally see in the literature (e.g. Forano, 2020)

Reply: We appreciate the reviewer for pointing to this important issue. We purposely planned to set the difference between the two force-fields weak, with the intention to make the association between the cue and the perturbation less consciously salient as possible (e.g. using magnitude difference or setting the difference between the conflicting force-field weak), to avoid any strong use of explicit strategies, either in the positive or negative way, for compensating the force.

However, following the reviewer's concern, we have now re-performed the opposing-force experiment by doubling the force-field strength, and have successfully obtained a

comparable level of compensation to the original experiment (Experiment 3-2). Please see Fig 3.

Regarding the point of whether learning of a weak force-field does not count as learning of the force. First, our aim is to see whether the participants can learn to differentiate the force producing pattern depending on the given context. Since we are consistently replicating the significant differentiation throughout the study with strong statistical effect size, the strength of the force-field may not be an issue. Second, it is not so uncommon to use force-fields $<10\text{N(m/s)}$, for example, Orban de Xivry, Shadmehr et al., 2011 uses gradually increasing force-field, and you can clearly see the signature of learning when the force is still weak. Similarly, Marko, Shadmehr et al., 2012 also uses force-field with the size smaller than 10N(m/s) . Therefore, weakness of the force-field itself is not necessarily the factor to make the learning easy and become a reason to undermine the result of the learning.

In any case, our observation that participants learned to produce the relevant amount (direction) of force depending on the given uncertainty level indicates that the brain indeed learned to compensate for the perturbation depending on the decision-uncertainty context.

Comment 23: Ln 522: This sentence does not flow and I don't understand what you are after.

Reply: Sorry for the unclear description. We have now revised the sentence as follows. Page 20, Lines 5-7.

“We examined whether the motor memory associated with the decision-uncertainty context is stimulus dependent or can generalize to other visual stimuli with matched decision uncertainty level.”

Comment 23: Ln 571: Learning coefficient based on a single point from the curve? That is far from standard. Does it matter whether you perform a regression without offset? I.e. more data into a single number.

Reply: Thank you very much for the suggestion. We have reanalysed and replaced all the data using the regression method and obtained the qualitatively and statistically equivalent results.

Comment 24: Ln 578: Ah, here is the retrieval ration. Last block, is how many trials?

Reply: One block consists of 72 trials with channel trials every 6 trials. Therefore, 12 trials in total. We now also provide the non-normalised force-compensation data in the Supplementary Materials Fig. S1.

Comment 25: Ln 580-593: You describe the model, but please also provide the fitting procedure and the values that came out of it. How did you construct the average curve?

Reply: We have provided further details on the fitting procedure and fitted parameters as suggested (Line 13-33). Supplementary Fig. S1 also show fitted parameters.

Comment 26: Ln 606: put 'expected difference ratio' in heading. Makes reading much easier. However, I also prefer a different naming here. It is not the predicted difference ratio, but the ration between the predicted and actual difference between conditions. A value of one indicates perfect performance. This paragraph is critical to understand many of the figures. This also makes me wonder, again, what I am looking at in figure 2E: if I understand your measure correctly, perfect performance would be 1.0, but you hardly touch 0.25. Can you really claim that they 'learn'? Same holds for figure 3.

Reply: Thank you very much for the suggestion. We have now decided to simply take a difference in force-compensation level of two different fields as an index of context dependent separation of motor memory.

We do understand the reviewer's concern. As written in the reply to your comment 3 & 15, we are confident that the participants indeed learned to separate the motor memory in a decision-uncertainty context dependent manner.

Reviewer #2

Comment 1: I find the study to be interesting and relevant to the field. The experimental manipulation is clever, simple and original. Overall, the experimental design and data analysis are sound.

Reply: Thank you very much for the comment. We appreciate the reviewer's positive evaluation about our paper!

Comment 2: 1. My major concern relates to the interpretation of the results. The authors suggest that their findings reflect a modulation of the decision process. However, I am not convinced that they provide sufficient evidence to support this hypothesis. The authors suggest that the visual cues used in the study are weak contextual cues, and thus unlikely to act as a tag on their own to encode different memories. However, the stimuli used to manipulate the level of uncertainty in which movements are planned are dynamic or imply motion, and such stimuli have been shown to tag different motor memories when the cued movement takes place within a time window of less than 600 ms (Howard et al., 2012; Howard et al., 2013).

Reply: Thank you very much for the clarification. The reviewer refers to Howard et al. (2013) to point out that the motion (dynamic) stimuli itself has been previously shown to be a weak contextual cue for motor learning. In their paper, however, the target rotating ‘*direction*’ was used as the cue to indicate the different force-field patterns (i.e. CW and CCW visual rotation each associated with different force-field). Similarly, they have also shown that prior cursor motion emulating the ‘follow-in’ hand movement path (which has been shown to be a contextual cue) prior to the action can tag the motor memory (Howard et al., 2012), but here again, there was a one-to-one fixed relationship between the motion ‘*direction*’ and the future perturbation type. Therefore, these study points to *direction* of the dynamic visual stimulus, not necessarily *dynamic* feature of the stimulus itself, can potentially be a contextual cue for motor memory.

In contrast, in our study, the motion ‘*coherence*’ level was used as the cue. Here, even if the participants moved to a particular direction according to their *direction* decision, the pattern of the force-field may differ depending on the motion coherence level. In other words, association between motion direction and the force-field pattern were unrelated by design. Therefore, to our understating, it is difficult to directly translate the result of Howard et al (2012, 2013) to our study, and that there is no prior evidence suggesting that motion (dynamic) stimuli itself, not the motion ‘*direction*’ cues, can be a contextual cue for motor memory. We have now clarified this point in the manuscript. Please see Page 5, lines 9-18.

Comment 3: I understand that Experiment 2.3 was designed to distinguish between the role of contextual uncertainty and the influence of stimulus features in producing the effects observed in Figures 1 and 2. However, a potential issue with this experimental design is that the cue and the resulting movement may not correspond, which could lead to reduced levels of sensorimotor tagging.

Reply:

Thank you very much for the comments.

The reviewer questions our control condition, stating that ‘*However, a potential issue with this experimental design is that the cue and the resulting movement may not correspond, which could lead to reduced levels of sensorimotor tagging.*’. We apologise if our claim was unclear. Correspondence between the cue ‘*direction*’ and the movement direction is irrelevant here, since the force-field in the main experiment is not associated with cue ‘*direction*’ but with the cue ‘*uncertainty*’. Even without such direction-action correspondence, the force-field was learned depending on the difference in decision-uncertainty level (Experiments 2-1, 3-1, 3-2, 4, 5).

The control condition in concern (original Experiment 2-4, now renamed as new Experiment 2-2) is designed to exclude the visual confound in Experiment 2-1. If the visual input of the motion with different *coherence* levels itself (100% and 3% coherent motion), not the uncertainty in the decision-making process, is sufficient to tag different motor memories (strong and weak force-fields), even excluding the explicit decision component

should lead to the same result. But the contextual effect was substantially weaker when the decision-process was excluded (new Experiment 2-2 and additional Experiment 3-3; Fig. 3E and Supplemental Fig. S2).

As the reviewer pointed out, the action direction and the motion direction can either match or mismatch in the control experiment. As mentioned above, however, motion direction and the force-field pattern have no relationship also in the main experiment, therefore, such motion-action direction coupling is not the sensorimotor tag of interest, and is not the feature which drove the result of the main experiments (Experiments 2-1, (new) 3-1, 3-2, 4, 5). Finally, by reanalysing the data of Experiment 2-2 and the new Experiment 3-3, we found no evidence that congruent (matched motion and action direction) situation had more facilitated learning compared to the incongruent situation (mismatch between the direction of motion and action). Taken together, we do not think sensorimotor confound, in terms of the match between the motion direction and action, is an issue in our control experiment. Please see Supplementary Materials Fig. S2.

Comment 4: Experiment 3 also uses a sequence of arrows which presumably implies motion, further suggesting the possibility of a sensorimotor confound. To test the hypothesis that actions can be tagged by the certainty/uncertainty of the decision context itself, a static and preferably abstract cue, irrelevant to movement execution, should be used to code the level of uncertainty. Alternatively, an interval longer than 600 ms between the presentation of the cue and movement initiation could be imposed to discard a sensorimotor confound.

Reply:

Here, the reviewer is raising two issues. One is about the ‘*sensorimotor confound*’ in the arrow sequence stimulus, another is the suggestion for an additional experiment.

Regarding the first point, same to the reply to your Comment 3, the match between the arrow *direction* and the action *direction* is not an issue here, because different force-field is associated depending on the uncertainty of the arrow direction (left-right arrow ratio of 90:10 or 45:55), not the direction of the stimulus. Therefore, the match between the arrow direction and the action direction is not a confound for the current force-field learning.

Reviewer’s comment ‘*a sequence of arrows which presumably implies motion*’ is not clear to us. For the arrow sequence stimulus, we present pictures of arrows (left and right) in a sequence, each for 2 frames followed by 2 frames blank. The only motion component embedded in the stimulus is the apparent motion induced by the transition from left (right) arrow picture to right (left) arrow picture. This motion component does not include the motion signal that specifies the direction, and since it will be less frequent in the certain condition (only twice for 20 pictures; thus kind of seeing a static arrow presented for 1.5 sec), it is rather a distractor for the decision in the uncertain condition. Our intention in Experiment 3 was to create a directional uncertainty signal different to random-dot-motion, to examine the visual modality independent component of decision uncertainty context.

Therefore, having a directional information in the arrow sequence was our intention, and we do not think the transfer is based on the visual similarity in the visual motion feature domain.

Nevertheless, whether an intuitive relationship between the cue and the action direction, e.g. left motion (arrow) indicating leftward action, facilitates the uncertainty context to be formed is very interesting. Therefore, as the reviewer requested, we performed an additional experiment to test whether the same pattern holds when using abstract cues, not motion or arrows. We first tried to manipulate the decision uncertainty level using static cues, such as changing the contrast level in a target detection task. However, it was difficult in our projector/lighting settings to stably manipulate the uncertainty level across participants, therefore, we conducted a same shape sequence task used in Experiment 4 but replacing the left and right arrows to triangles and squares. Participants were asked to judge which of the two shapes were more frequently presented in the sequence and then moved to the direction (left or right) where the target shape is displayed (see below; Fig R1). The shape-direction relationship was fixed (i.e. triangle displayed on the left, square on the right) within each participant, and counter-balanced across participants. Uncertainty level was manipulated by changing the ratio of the appearance of the shapes (ratio of 90:10 or 45:55). Depending on the uncertainty level, we assigned either CW or CCW force-field directions. Strength of the force-field increased in a same manner as in Experiment 2-5, which the final viscosity level was ± 5 (N/[ms⁻¹]).

We found significant difference in force compensation level between the certain and uncertain conditions (paired t-test, $t[13]=2.64$, $p=0.02$, $d_z=0.7$), replicating our present findings (Fig R1 below). However, the amount of compensation was weaker than the original experiments using motion stimulus, understandably due to the non-intuitive decision-action mapping. Arbitrary cue-action transformation has been shown to require qualitatively different neuronal computation compared to the transformation using simple spatial cues (Murray and Wise, 2000, McDougle et al., 2022), such as involving the medial temporal and prefrontal cortices. Thus, the decision-uncertainty context may be facilitated if the decision can be fluently transformed into action without additional computation, such as with the intuitive, well-learned cues. Further training to establish the shape-direction association may improve the contextualisation of motor memory, but we feel that is in a different scope of the present series of studies. Putting this result in the main text may blur the focus of the paper, therefore, we have decided not to include the data in the main manuscript, but present it here.

It is difficult to completely dissect/remove the sensorimotor component from the cascade of decision-action transformation process (Gold and Sadlen 2007, Cisek and Kalaska, 2010). Rather, we believe that the difference in the neural activity pattern of decision-action transformation is the tag for the motor memory, which we have demonstrated in our new Experiment 5. We have made these points clear in the revised manuscript. Please see Fig. 5 and paragraph in Page 13, starting from lines 34.

Taken together, we don't think using motion or arrow for the decision stimuli, will limit our findings

Fig. R1. Decision uncertainty cue using arbitrary shapes.

A: Participants were asked to judge which of the two shapes were more frequently presented in the sequence and then moved to the direction (left or right) where the target shape is displayed. The shape-direction relationship was fixed (i.e. triangle displayed on the left, square on the right) within each participant, and counter-balanced across participants. Uncertainty level was manipulated by changing the ratio of the appearance of the shapes (ratio of 90:10 or 45:55). Depending on the uncertainty level, we assigned either CW or CCW force-field directions. Strength of the force-field gradually increased in a same manner as in Experiment 3-2, which the final viscosity level was ± 5 (N/[ms⁻¹]).

B, C: Result of the learning. Participants gradually increased the force compensation level towards the end, resulting in significant difference in force compensation level between the two different uncertainty levels. However, the effect was 20% of that observed during motion signal, indicating that the contextual effect of decision uncertainty is attenuated when the fluent decision-action transformation is prevented.

Shade indicate the standard error of means across participants. For the boxplots, each dot represents each participant, and the midline of the box represents the median of the data. The box itself spans from the 25th to 75th percentile, and the whiskers show the range (min to max) of the data. Outliers are determined by data points that are more than [whisker length x size of box from the box]. *: $p < 0.05$.

Comment 5: The figures in the manuscript require additional work to ensure that readers can understand them by reading the captions alone. In many instances, the information provided in the figure does not match the descriptions in the Results or Methods section. For example, Figure 1.B suggests that only the retrieval phase includes error-clamp trials, while the learning phase does not. Similarly, Figure 1.C specifies probe trials only during the retrieval phase, which contradicts the information provided in the Methods. To clarify the experimental design, it would be helpful to merge Figure 1.A and Supplementary Figure 1 so that readers can see both the

display and the experimental setting. Additionally, in Figure 3, the cue changes, but the display is not the same as in Figure 1, even though the text suggests it should be. To address this, it may be useful to include a replica of Figure 1.A, but with an arrow replacing the dots. Please clarify in the caption of Figure 1 what the error bars indicate Overall, the figures would benefit from additional effort to ensure consistency and clarity.

Comment 6: The description of Experiment 1 lacks clarity, which makes it difficult for the reader to understand the methodology and the results. The information provided in the Results section is incomplete and inconsistent with what is presented in Figure 1. Although the caption mentions two phases of the experiment, "learning" and "retrieval," the text fails to elaborate on them adequately. The figure suggests that the retrieval phase solely consists of probe trials, but the Methods section states that the retrieval phase is the same as the learning phase except for the type of probe trials. Additionally, the Methods section lacks important details, such as whether there were any breaks within each phase and if so, their duration and location. The Methods section also does not specify whether there were any breaks between the learning and retrieval phases or how long it took to complete the entire task.

Reply: We appreciate the reviewer for the detailed suggestion and apologise for the lack of detail in the previous version. Now all of the Figures are revised accordingly, together with the text in the manuscript referring to them. Duration of the experiments has been also added. There was a short break (<1min) in between the blocks, but there was no special break in between the learning and the test phase. This structure was same for all of the experiments. Please refer to Figures and the Method section Page 17 lines 25-29.

Comment 6: have a concern regarding the distinction between the learning and retrieval phases in this study. According to the Methods, the only difference between these phases is the type of error-clamp trials used. However, it is not clear why the second phase is labeled as "retrieval." Figure 1D suggests that subjects have already reached asymptotic performance by the end of the learning phase. If there is no break between the two phases, then the so-called "retrieval" phase could be interpreted as overlearning, i.e., continued training at the asymptote. Additionally, I am unclear why only the probe trials from the retrieval phase are considered to measure memory retention/retrieval. How do the two phases differ beyond asymptotic performance? If the goal was to measure memory retention/retrieval after learning, why not conduct a separate test session after asymptotic performance was reached, without further learning?

Reply: Thank you very much for pointing this out, and we apologise for the confusion caused by the naming of the conditions.

The goal of Experiment 1 is to examine how the content of learning following a certain (uncertain) motion decision transfer across different uncertainty levels. For this purpose, it is necessary for the participants to maintain the learned level throughout the retrieval phase (now, named as test phase). Previous literature has shown that the learning decays during the error-clamp probe trials (Scheidt et al., 1997), therefore, we ‘topped-up’ the learning in-

between the probe trials with the force-field trials. Note that, error-clamp trials in the retrieval phase included different motion coherence levels, but the force-field trials only included the coherence level they experienced in the learning phase (100% or 3% depending on their group). This prevented any additional motor learning to occur for other motion coherence level. Therefore, the probe trials were the only trials we can evaluate the transfer level.

We agree with the reviewer that using the word ‘retrieval’ may confuse the reader, therefore, we have now replaced it with the name ‘test phase’. Also, we have revised the Method of Experiment 1 to make the experimental conditions clear.

Comment 7: The Discussion could be extended to elaborate on how this study differs from previous literature addressing the impact of context on motor memory tagging. The work by Song and collaborators (2013, 2015) showing that visuomotor adaptation memories acquired in a context of attentional distraction are better recalled under a similar level of distraction than under a no-distraction context, is very relevant to this study but is not mentioned neither in the Introduction nor in the Discussion. The authors could also consider whether similar mechanisms may be at play in the motor system's use of different strategies for motor control in the face of uncertainty, as suggested in previous literature (e.g., Chouinard and Paus, 2005).

Reply: We appreciate the reviewer for pointing to the relevant literatures. Now, they are incorporated and discussed in the discussion section. Also, we have discussed the novelty of our finding in relation to the contextual inference models, such as the COIN model, as suggested by reviewer 1. Please see the discussion section. Page 13, Lines 9-23, and Page 14, lines 3-11.

Comment 8: Finally, I have some minor comments. Please clarify whether "nonerror-clamp trials" refer to regular learning trials in which no clamp is applied (line 432). Lines 129-141 could be reworded for clarification.

Reply: Yes, nonerror-clamp trials are the trials which the participants experience the force-field. Now, we have re-named it as “force-field trials” throughout the manuscript. Thanks for the suggestion.

Reviewer #3

Comment 1: Overall, this is a decent amount of work, and the 3 experiments (and the controls) are logically linked and address a related hypothesis. However, there are several shortcomings that should be addressed in detail. In addition, the speculation that planning-related mechanism underlies the covert internal decision is a testable question and my suggestion is to perform this experiment to strengthen the main finding (see my comments below).

Reply: Thank you very much for the comment. We appreciate the reviewer's positive evaluation of our paper, and the constructive suggestion to support our claim.

Comment 2: There is inconsistency in the amount of learning across the different experiments. To this reviewer it is not clear why participants reached different learning levels across the different experiments. For example, in exp. 1, participants reached a level of ~65-70% of the ideal force in the different uncertainty conditions, while in the same condition (e.g., strong FF and certain) in exp. 2-1, participants reached at maximum 55%, and ~50% in exp. 3 for the same condition. At first, I thought that the learning phase was shorter in these experiments but when we look at the late phase of the learning curve (Fig., 2C), it looks like the participants reached asymptote, suggesting that something else beyond the length of the learning phase, affected the learning process. Do the authors have any explanation for what might cause this? Also, I am curious as to whether some interference could have happened during the switch between the FFs.

Reply: We apologise if the logic was unclear. In Experiment 1, participants learned only one condition, namely, force-field learning following either certain (100% coherent motion) or uncertain (3% coherent motion) visual stimuli. In contrast, in Experiment 2, the different magnitude of force-fields, each associated with different decision uncertainty level, is learned simultaneously. If the participants cannot contextualise different force-fields with different decision uncertainties, the optimal way to deal with this situation is to minimise the error by producing the force which is in-between the two magnitude levels. In contrast, any difference in magnitude levels between the two uncertainty conditions will point to the context-based separation of motor memory.

Thus, the difference of compensation level between Experiment 1 (certain condition) and Experiment 2-1 (strong force condition) is simply due to whether there is an interference from the other force-field condition (Experiment 2) or not (Experiment 1).

The difference between Experiment 2 and 3 is due to the slight difference in design. In Experiment 2, we adopted a conventional reaction time paradigm for the motion direction decision, in which whenever the participant detected the direction, they can initiate the action. Therefore, the stimulus duration of 100% and 3% coherent motion differed. In Experiment 3, however, to match the duration of the motion stimuli with the arrow-sequence stimuli, the random-dot motion duration was fixed to 1500ms for both 100% and 3% coherent motion. Making the duration equal may have slightly weakened the result, possibly attenuating the influence of decision uncertainty on action, nevertheless, we still found reliably strong signature of contextualization of motor memory.

Taken together, difference in the compensation level across experiments arose from the difference in the detailed design of the experiments, which does not affect our conclusion.

Comment 3: The quantification of the learning using the learning coefficient based on a single point of the force profile (at max velocity) is unclear to this reviewer and seems to be very sensitive to outliers in the data. Typically, to give a robust measure of learning, the actual forces in the channel's trials should be regressed with the ideal force, and then the regression coefficient is used to determine how much the participant learns from the ideal FF. This robust regression is immune to any unstable movements that might occur. I suggest conducting the analysis of learning, but instead of focusing on a single data point, perform the regression across the entire force profile. Related, the paper is largely lacking information about the force profiles across learning and retrieval. This should be at least added to the Supplementary Information.

Reply: Thank you very much for the suggestions. We re-analysed and replaced all the data based on the regression, and all the figures are re-created using the % compensation level. Qualitatively (statistically), there was no marked difference compared to our previous analysis.

In the originally presented experiments, we did not have enough channel (error-clamp) trials to evaluate the time course of the learning during the learning phase. However, now, the channel trials are included in the new experiments we performed for the revision (new Experiment 3-2, 3-3, and 5), thus is now presented in the main text and in the supplementary materials. Please refer to Fig.3 and Supplemental Fig. S2.

Comment 4: The speed of movement across the different conditions is lacking in all experiments. This analysis should be carefully conducted, and the results should appear in the manuscript. The speed of a movement can hint to the learning mechanism during the learning phase (i.e., feedforward vs. feedback). The fact that the authors analyzed the force profile at a single point and only during the force channel trials do not provide deep understanding of the nature of the learning.

Reply: Thank you very much for pointing this out. As suggested, we have now added the analysis of peak velocity and the description related to this issue. Please see Page 5 Lines 19-25 Page 7 lines 33-38 and Supplementary Fig S1.

In short, in Experiment 1, the pattern of movement velocity across different motion coherence levels differed across different groups (Certain or Uncertain). In Certain group, velocity for the 100% coherent motion was faster than the other coherence levels. In contrast, such gradation diminished for the Uncertain group.

In Experiment 1, participants were continuously exposed to one type of coherent motion (3% or 100%) in the learning phase, and more frequently than the other also in the test phase (because the force-field trials only had one uncertainty level, which they have learned). This may have increased the movement velocity in the trials with corresponding motion coherence level. Supporting this interpretation, velocity difference was not observed in the dual force-field conditions (Experiment 2, 3, 4, 5), where the participants were exposed to two different types of stimuli with different uncertainty level for the same amount.

Therefore, difference in the design may have led to this slight discrepancy of the result across experiments. Nevertheless, only finding the velocity different in Experiment 1 and not in other experiments indicates that the velocity is not a strong denominator for the context-dependent learning using decision-uncertainty.

Comment 5: The speculation that uncertainty of the decision forms a context that might affect premovement (e.g., planning) is very interesting, but direct evidence in this paper is lacking. Nevertheless, since this is a testable prediction using a rather simple behavioral paradigm (see for example the follow-through experiments of opposing FFs in previous work), I highly recommend performing this experiment and contrasting planning vs. execution while manipulating the uncertainty of the decision. This would confirm the main conclusion, and most importantly, confirm the speculated mechanism.

Reply: Thank you very much for suggesting this important experiment. As suggested, we performed the additional experiment. Please see detail in Experiment 5 (Fig. 5).

As in Howard et al. (2015), Sheahan et al., (2016), participants made a follow-through reaching movement, in which they first made a straight movement to a central target and then immediately reached to the left/right secondary target. Instead of pre-defining the secondary target position as has been done in the previous studies, we showed a certain (100% coherence level) or uncertain (3%) random-dot motion for 1000ms. Participants decided the motion direction (left or right) within this 1000ms, and immediately after the disappearance of the motion stimulus, they made a follow-through reaching movement, terminating at the secondary target of the decided motion direction. Conflicting force-fields (CW or CCW) were applied only to the first common path to the central target depending on the uncertainty of the motion stimuli (Fig. 5A, B).

We found that participants can learn two conflicting forces depending on the two different uncertainty level of the decision stimulus. Moreover, such learning transferred to the trials where the participants were asked to stop at the first target during the movement (NOGO trials). This indicates that the plan of the decided action is sufficient to retrieve the motor memory.

Taken together, this result suggests that the decision uncertainty is contextualising the action already at the planning stage. We thank the reviewer again for suggesting this important experiment to strengthen our result.

Comment 6: Generalization of learning from stronger to weaker FF (and perhaps verse versa) is highly possible. If this is the case, then the amount of learning reported when changing the context from high to low uncertainty (or the opposite) might be also related to the generalization between the two environments. This should be somehow quantified and further discussed.

Reply: The reviewer writes ‘Generalization of learning from stronger to weaker FF (and perhaps verse versa) is highly possible. If this is the case, then the amount of learning reported when

changing the context from high to low uncertainty (or the opposite) might be also related to the generalization between the two environments.' We are not sure which data you are referring to. Change in contexts from higher to lower uncertainty occurs in Experiment 1, but here, participants are only exposed to a single level of force-field magnitude. In Experiment 2, participants learned two different magnitudes of force-field each associated with different decision uncertainty levels, but the context did not change. We assume that you are asking us to discuss about the possible generalisation of the learning between strong and weak force-field in Experiment 2. Please let us know if we have any misunderstanding about your point.

First, in the magnitude experiment (Experiment 2 and 3), since the perturbation is applied from the same direction for both strong and weak conditions, participants will learn to produce the force in the same direction. The question here is, whether the participants can appropriately produce the relevant force magnitude depending on the assigned decision uncertainty context.

There could be two reasons which would prevent us to observe such context dependent learning. One possibility is that the decision uncertainty does not work as a context. In such case, participants will produce equal amount of force, which would be somewhere in between the two required force, to minimise the overall error. Another possibility is that, as the reviewer points out, generalisation of learning across different magnitude level occurs. If the generalisation function for each force magnitude is overlapping, the two different outputs will be indistinguishable and as a result, the difference in force output would become smaller (towards the direction to diminish the context dependent learning).

Therefore, we do not deny the existence of generalisation, however, such factor would work in the direction to hide the context dependent learning. The fact that we were able to observe the decision-uncertainty dependent motor learning shows that the effect of generalisation was insufficient to cover the effect.

Comment 7: The timing of the decision process might also affect the learning. It is not clear (perhaps the reviewer missed it) if there is a correlation between the decision time (time from dot's motion onset to movement onset) and the learning amount and whether there were differences between the different uncertainty conditions/groups.

Reply: Thank you very much for the clarification. Indeed, in Experiment 1 and 2, the duration of the visual stimulus was shorter for the certain compared to the uncertain condition (i.e. reaction time paradigm of perceptual decision making.). However, in Experiment 2-4 (now, renamed as Experiment 2-3), we showed that such effect was significantly smaller than that observed in Experiment 2-1. Also, in Experiment 4 and 5, where the stimulus duration was set equal across different decision uncertainty contexts, we still observed a significant effect of contextual learning. Finally, as the reviewer suggested, when we additionally performed a correlation analysis, we did not find any systematic relationship between the reaction time and the contextual separation of motor memory.

Taken all together, duration of the stimulus is not the main contributor for separating different contexts. This is now described in Page 7, lines 14-19.

Comment 8: Secondary comment: Fig. 1E. it is very puzzling as to why participants in the Uncertain condition would learn less if they are exposed to a 100% certainty of where to move. How was this expressed? Is it in a form of lower force profile compared to learning uncertain condition? Also, I am curious about comparing the retrieval ratio across groups but here matching the uncertainty during learning. That is the retrieval for the uncertain group during 100% and the certain group in the 3% uncertainty. This would tell us whether the retrieval is purely context dependent or there is something else occurring during the decision process.

Reply: Thanks for the comment. In Figure 1E, the vertical axis is the force-compensation level in the test phase divided by the level participants have reached at the end of learning phase (end of Figure 1D), which indicates the amount of motor memory retained in the test phase compared to the end of learning phase.

As you have suggested (please see above comment 3), we have re-calculated the force compensation level by the regression method, and normalised the data of the test phase with that of end of the learning phase. Similar to the previous results, we still find clear reduction of force compensation level in the 100% coherence level compared to the 3% coherence level, which indicates the context-dependent motor learning. Force compensation level before normalisation is now provided in the Supplemental Fig. S1.

During the learning phase, uncertain-decision group was exposed only to 3% coherent motion, and certain-decision group to 100% coherent motion. This is to avoid any exposure to other uncertainty levels during the learning, to purely evaluate the generalisation of learning to other uncertainty levels in the test phase. Therefore, we cannot compare the result of matched uncertainty levels between the groups during the learning.

We have now revised the description of Experiment 1 and the Fig.1 and the associated figure legend to make these point clearer.

Decision Letter, first revision:

8th April 2024

Dear Dr. Hagura,

Thank you for your patience as we've prepared the guidelines for final submission of your Nature Human Behaviour manuscript, "Decision uncertainty as a context for motor memory" (NATHUMBEHAV-23041060A). Please carefully follow the step-by-step instructions provided in the attached file, and add a response in each row of the table to indicate the changes that you have made. Please also address the additional marked-up edits we have proposed within the reporting summary. Ensuring that each point is addressed will help to ensure that your revised manuscript can

be swiftly handed over to our production team.

We would hope to receive your revised paper, with all of the requested files and forms within two-three weeks. Please get in contact with us if you anticipate delays.

Nature Human Behaviour offers a Transparent Peer Review option for new original research manuscripts submitted after December 1st, 2019. As part of this initiative, we encourage our authors to support increased transparency into the peer review process by agreeing to have the reviewer comments, author rebuttal letters, and editorial decision letters published as a Supplementary item. When you submit your final files please clearly state in your cover letter whether or not you would like to participate in this initiative. Please note that failure to state your preference will result in delays in accepting your manuscript for publication.

In recognition of the time and expertise our reviewers provide to Nature Human Behaviour's editorial process, we would like to formally acknowledge their contribution to the external peer review of your manuscript entitled "Decision uncertainty as a context for motor memory". For those reviewers who give their assent, we will be publishing their names alongside the published article.

Cover suggestions

We welcome submissions of artwork for consideration for our cover. For more information, please see our guide for cover artwork.

ORCID

Non-corresponding authors do not have to link their ORCIDs but are encouraged to do so. Please note that it will not be possible to add/modify ORCIDs at proof. Thus, please let your co-authors know that if they wish to have their ORCID added to the paper they must follow the procedure described in the following link prior to acceptance: <https://www.springernature.com/gp/researchers/orcid/orcid-for-nature-research>

Nature Human Behaviour has now transitioned to a unified Rights Collection system which will allow our Author Services team to quickly and easily collect the rights and permissions required to publish

your work. Approximately 10 days after your paper is formally accepted, you will receive an email in providing you with a link to complete the grant of rights. If your paper is eligible for Open Access, our Author Services team will also be in touch regarding any additional information that may be required to arrange payment for your article.

Please note that *Nature Human Behaviour* is a Transformative Journal (TJ). Authors may publish their research with us through the traditional subscription access route or make their paper immediately open access through payment of an article-processing charge (APC). Authors will not be required to make a final decision about access to their article until it has been accepted. Find out more about Transformative Journals

Please use the following link for uploading these materials:
[REDACTED]

Best regards,
[REDACTED]

On behalf of
[REDACTED]

Reviewer #1:
Remarks to the Author:
Decision uncertainty as a context for motor memory

The authors have done a great job, running additional experiments to further confirm their findings presented in the first version of this paper. They have addressed all my previous concerns and incorporated the requests about discussing the meaning and interpretation of their findings. My only remaining problem is that the newly written pieces of text need editing. The content of the sentences is clear, but the writing does not match the level of the first submission. I highlight some problems below, but please have the manuscript read by a writing expert.

Minor comments:

Ln 52: I would write this in plural ("processes" and "specify").

Ln 89 "a same decision making task", rephrase.

Ln 100: 'the' certain-decision group

Ln 109: "The decision stimuli in the force-filled trials were the same as in the learning phase" – check writing in many places. This was just one sentence where words are missing.

Ln 114: not 'a same', but 'the same' or 'a similar'.

Ln 161: Previous study showed -> incorrect grammar.

Ln 168: I believe I get the point, but this sentence is illegible.

Fig 2 and accompanying text. Indicate which directions participants reached and that RDM stimuli switched both direction and certainty.

Ln 239: Reference to Fig 2B seems wrong.

Ln 378: 'a straight reaching', not 'reaching'.

Ln 425: writing of sentence is bad

Ln 812: I am a bit confused. Position does not go into the equation, does it?

Kind regards,

Luc

Reviewer #2:

Remarks to the Author:

The authors have addressed my concerns.

I suggest they put additional effort into revising the English throughout the manuscript. Many of the additions to the original manuscript are poorly written and, as a result, not sufficiently clear.

Reviewer #3:

Remarks to the Author:

The authors have thoroughly addressed my comments as well as those from the other reviewers. I have only a minor suggestion remaining (if space permits), which I believe could further enhance the manuscript.

It would be advantageous to align the current findings with the motor control community's recent insights into the interplay between explicit and implicit processes during sensorimotor learning. Specifically, examining whether the current findings regarding "decision uncertainty as a context" can be linked to either process. Incorporating this perspective could deepen the discussion by placing your results within the expansive framework of sensorimotor learning.

For example, a recent study has used the opposing perturbations paradigm of follow-through movements (ref #7, 18) to explore the contribution of explicit vs. implicit process in this type of motor learning (e.g., Dawidowicz, G. et al.,(2022). Separation of multiple motor memories through implicit and explicit processes. *Journal of Neurophysiology*, 127(2), pp.329-340).

A brief discussion, about the potential explanation (or even speculations) of how the current study could be related to the implicit-explicit topic, would be helpful for the readers.

Author Rebuttal, first revision:

POINT-BY-POINT REPLY LETTER

Revised sections are highlighted in red in the manuscript.

Reviewer #1

Comment:

The authors have done a great job, running additional experiments to further confirm their findings presented in the first version of this paper. They have addressed all my previous concerns and incorporated the requests about discussing the meaning and interpretation of their findings. My only remaining problem is that the newly written pieces of text need editing. The content of the sentences is clear, but the writing does not match the level of the first submission. I highlight some problems below, but please have the manuscript read by a writing expert.

Reply: We are pleased to hear that we were able to address your concern! We would again like to express our appreciation regarding your constructive and detailed comments, which have significantly improved our manuscript.

We apologise for the quality of the English. The manuscript has now been edited by a professional editing service. We hope that the manuscript is now more readable.

Reviewer #2 (Remarks to the Author):

Comment:

The authors have addressed my concerns.

I suggest they put additional effort into revising the English throughout the manuscript. Many of the additions to the original manuscript are poorly written and, as a result, not sufficiently clear.

Reply: We are very glad to hear your positive evaluation of our revised manuscript! We apologize for the writing quality. The paper has now been edited by a professional proof editing service. We hope this will make the overall logic of the manuscript clearer.

Reviewer #3 (Remarks to the Author):

Comment :

The authors have thoroughly addressed my comments as well as those from the other reviewers. I have only a minor suggestion remaining (if space permits), which I believe could further enhance the manuscript.

It would be advantageous to align the current findings with the motor control community's recent insights into the interplay between explicit and implicit processes during sensorimotor learning. Specifically, examining whether the current findings regarding "decision uncertainty as a context" can be linked to either process. Incorporating this perspective could deepen the discussion by placing your results within the expansive framework of sensorimotor learning.

For example, a recent study has used the opposing perturbations paradigm of follow-through movements (ref #7, 18) to explore the contribution of explicit vs. implicit process in this type of motor learning (e.g., Dawidowicz, G. et al.,(2022). Separation of multiple motor memories through implicit and explicit processes. Journal of Neurophysiology, 127(2), pp.329-340).

A brief discussion, about the potential explanation (or even speculations) of how the current study could be related to the implicit-explicit topic, would be helpful for the readers.

Reply: We are very happy to hear that we were able to address your concerns! Thank you very much for making us aware of the relation between the explicit/implicit component of learning to our study. Indeed, we agree that this is one of the current topics in the field of motor learning, however, since we did not explicitly examine the contribution of these two processes in the study, our ability to discuss this point is rather limited.

Speculatively, however, we believe that the contextual learning based on decision-uncertainty in this study is based on implicit learning for two reasons. One, we use force-field adaptation for the task, where the effect of explicit-strategy on motor learning is known to be limited (Schween et al., 2020) compared to the visuomotor adaptation. Second, in two of the experiments (Exp. 2 and 4), the participants were able to separate strong and weak force-fields according to the decision-uncertainty context. Since the two perturbations are both from the same direction, it would be very demanding to change the magnitude of the resisting force explicitly/consciously depending on the preceding uncertainty of the decision.

Therefore, we believe that the major contributor of the learning is the implicit component of learning.

We have now briefly touched on this point in the discussion section, page 11, lines 19-34.

Final Decision Letter:

Dear Dr Hagura,

We are pleased to inform you that your Article "Decision uncertainty as a context for motor memory", has now been accepted for publication in *Nature Human Behaviour*.

Please note that *Nature Human Behaviour* is a Transformative Journal (TJ). Authors may publish their research with us through the traditional subscription access route or make their paper immediately open access through payment of an article-processing charge (APC). Authors will not be required to make a final decision about access to their article until it has been accepted. Find out more about Transformative Journals

With best regards,

[REDACTED]